# Neural Latent Geometry Search: Product Manifold Inference via Gromov-Hausdorff-Informed Bayesian Optimization

**Haitz Sáez de Ocáriz Borde***
Oxford Robotics Institute
University of Oxford

**Álvaro Arroyo***
Oxford-Man Institute
University of Oxford

**Ismael Morales López**
Mathematical Institute
University of Oxford

**Ingmar Posner**
Oxford Robotics Institute
University of Oxford

**Xiaowen Dong**
Machine Learning Research Group
University of Oxford

## Abstract

Recent research indicates that the performance of machine learning models can be improved by aligning the geometry of the latent space with the underlying data structure. Rather than relying solely on Euclidean space, researchers have proposed using hyperbolic and spherical spaces with constant curvature, or combinations thereof, to better model the latent space and enhance model performance. However, little attention has been given to the problem of automatically identifying the optimal latent geometry for the downstream task. We mathematically define this novel formulation and coin it as neural latent geometry search (NLGS). More specifically, we introduce an initial attempt to search for a latent geometry composed of a product of constant curvature model spaces with a small number of query evaluations, under some simplifying assumptions. To accomplish this, we propose a novel notion of distance between candidate latent geometries based on the Gromov-Hausdorff distance from metric geometry. In order to compute the Gromov-Hausdorff distance, we introduce a mapping function that enables the comparison of different manifolds by embedding them in a common high-dimensional ambient space. We then design a graph search space based on the notion of *smoothness* between latent geometries, and employ the calculated distances as an additional inductive bias. Finally, we use Bayesian optimization to search for the optimal latent geometry in a query-efficient manner. This is a general method which can be applied to search for the optimal latent geometry for a variety of models and downstream tasks. We perform experiments on synthetic and real-world datasets to identify the optimal latent geometry for multiple machine learning problems.

## 1 Introduction

There has been a recent surge of research employing ideas from differential geometry and topology to improve the performance of learning algorithms [Bortoli et al., 2022, Hensel et al., 2021, Chamberlain et al., 2021, Huang et al., 2022, Barbero et al., 2022a,b]. Traditionally, Euclidean spaces have been the preferred choice to model the geometry of latent spaces in the ML community [Weber, 2019, Bronstein et al., 2021]. However, recent work has found that representing the latent space with a geometry that better matches the structure of the data can provide significant performance enhancements in

---

*Equal contribution

37th Conference on Neural Information Processing Systems (NeurIPS 2023).

both reconstruction and other downstream tasks [Shukla et al., 2018]. In particular, most works have employed *constant curvature model spaces* such as the Poincaré ball model [Mathieu et al., 2019], the hyperboloid [Chami et al., 2019], or the hypersphere [Zhao et al., 2019], to encode latent representations of data in a relatively simple and computationally tractable way.

While individual model spaces have sometimes shown superior performance when compared to their Euclidean counterparts, more recent works [Gu et al., 2018, Skopek et al., 2019, Sáez de Ocáriz Borde et al., 2023b,a, Zhang et al., 2020, Fumero et al., 2021, Pfau et al., 2020] have leveraged the notion of *product spaces* (also known as *product manifolds*) to model the latent space. This idea allows to generate more complex representations of the latent space for improved performance by taking Cartesian products of model spaces, while retaining the computational tractability of mathematical objects such as *exponential maps* or *geodesic distances*, see Sáez de Ocáriz Borde et al. [2023b]. Despite the success of this methodology, there exists no principled way of obtaining the *product manifold signature* (*i.e.*, the choice and number of manifold components used to generate the product manifold and their respective dimensionalities) to optimally represent the data for downstream task performance. This procedure is typically performed heuristically, often involving a random search over the discrete combinatorial space of all possible combinations of product manifold signatures, which is an exceedingly large search space that hampers computational efficiency and practical applicability. Some other work related to latent space geometry modelling can be found in Lubold et al. [2023], Hauberg et al. [2012], Arvanitidis et al. [2017].

**Contributions. 1)** In this paper, we consider a novel problem setting where we aim to search for an optimal latent geometry that best suits the model and downstream task. As a particular instance of this setting, we consider searching for the optimal product manifold signature. Due to the conceptual similarity with neural architecture search (NAS) strategies [Elsken et al., 2018, Zoph and Le, 2016, Pham et al., 2018], we coin this problem as *neural latent geometry search (NLGS)*, which we hope will encourage additional work in the direction of optimal latent geometry inference. We test our framework on a variety of use cases, such as autoencoder reconstruction [Mathieu et al., 2019] and latent graph inference [Sáez de Ocáriz Borde et al., 2023b], for which we create a set of custom datasets.

**2)** To search for the product manifold signature, we must be able to compare product manifolds. This is traditionally done by computing the Hausdorff distance between manifolds [Taha and Hanbury, 2015], which however requires manifolds to reside in the same metric space. To address this limitation, in this work we develop to our knowledge the first computational method to mathematically compare product manifolds. Our approach generalizes classical algorithms and allows the comparison of manifolds existing in different spaces. This is achieved by defining an isometric embedding that maps the manifolds to a common high-dimensional ambient space, which enables the computation of their Gromov-Hausdorff distances.

**3)** Leveraging the Gromov-Hausdorff distances between candidate latent space manifolds, we design a principled and query-efficient framework to search for an optimal latent geometry, in the sense that it yields the best performance with respect to a given machine learning model and downstream task.

Our approach consists of constructing a geometry-informed graph search space where each node in the graph represents a unique candidate product manifold, associated with the model performance using this manifold as its embedding space. The strength of edges in the graph are based on the inverse of the Gromov-Hausdorff distance, thereby encoding a notion of "closeness" between manifolds in the search space. We then perform efficient search over this space using Bayesian optimization (BO). We compare our proposed method with other search algorithms that lack the topological prior inherent in our model. Empirical results demonstrate that our method outperforms the baselines by a significant margin in finding the optimal product manifold.

**Outline.** In Section 2 we discuss manifold learning, and product manifolds of constant curvature model spaces such as the Euclidean plane, the hyperboloid, and the hypersphere. We also review relevant mathematical concepts, particularly the Hausdorff and Gromov-Hausdorff distances from metric geometry [Gopal et al., 2020]. Section 3 presents the problem formulation, the proposed methodology to compare product manifolds, as well as how the search space over which to perform geometry-informed Bayesian optimization is constructed. Finally, Section 4 explains how our custom

synthetic and real-world datasets were obtained, and the empirical results. Lastly, in Section 5 we conclude and discuss avenues for future work.[2]

# 2 Background

**Manifold Learning** Manifold learning is a sub-field of machine learning that uses tools from differential geometry to model high-dimensional datasets by mapping them to a low-dimensional latent space. This allows researchers to analyze the underlying structure of data and improve machine learning models by capturing the geometry of the data more accurately. Manifold learning is based on the assumption that most observed data can be encoded within a low-dimensional manifold (see Figure 1) embedded in a high-dimensional space [Fefferman et al., 2013]. This has seen applications in dimensionality reduction [Roweis and Saul, 2000, Tenenbaum et al., 2000], generative models [Goodfellow et al., 2014, Du et al., 2021, Bortoli et al., 2022], and graph structure learning for graph neural networks (GNNs) [Topping et al., 2021]. In all these application, the key is to find a topological representation as an abstract encoding that describes the data optimally for the downstream task.

**Product Manifolds.** In this work, we model the geometry using model space Riemannian manifolds (Appendix A.1) and Cartesian products of such manifolds. The three so-called model spaces with constant curvature are the Euclidean plane, $\mathbb{E}^n = \mathbb{E}_{K_{\mathbb{E}}}^{d_{\mathbb{E}}} = \mathbb{R}^{d_{\mathbb{E}}}$, where the curvature $K_{\mathbb{E}} = 0$; the hyperboloid, $\mathbb{H}^n = \mathbb{H}_{K_{\mathbb{H}}}^{d_{\mathbb{H}}} = \{\mathbf{x}_p \in \mathbb{R}^{d_{\mathbb{H}}+1} : \langle \mathbf{x}_p, \mathbf{x}_p \rangle_{\mathcal{L}} = 1/K_{\mathbb{H}}\}$, where $K_{\mathbb{H}} < 0$ and $\langle \cdot, \cdot \rangle_{\mathcal{L}}$ is the Lorentz inner product; and the hypersphere, $\mathbb{S}^n = \mathbb{S}_{K_{\mathbb{S}}}^{d_{\mathbb{S}}} = \{\mathbf{x}_p \in \mathbb{R}^{d_{\mathbb{S}}+1} : \langle \mathbf{x}_p, \mathbf{x}_p \rangle_2 = 1/K_{\mathbb{S}}\}$, where $K_{\mathbb{S}} > 0$ and $\langle \cdot, \cdot \rangle_2$ is the standard Euclidean inner product. These have associated exponential maps and distance functions with closed form solutions, which can be found in Appendix A.2. A product manifold can be constructed using the Cartesian product $\mathcal{P} = \times_{i=1}^{n_{\mathcal{P}}} \mathcal{M}_{K_i}^{d_i}$ of $n_{\mathcal{P}}$ manifolds with curvature $K_i$ and dimensionality $d_i$. Note that both $n_{\mathcal{P}}$ and $d_i$ are hyperparameters that define the product manifold $\mathcal{P}$ and that must be set a priori. On the other hand, the curvature of each model space $K_i$ can be learned via gradient descent. One must note that the product manifold construction makes it possible to generate more complex embedding spaces than the original constant curvature model spaces, but it does not allow to generate any arbitrary manifold nor to control local curvature.

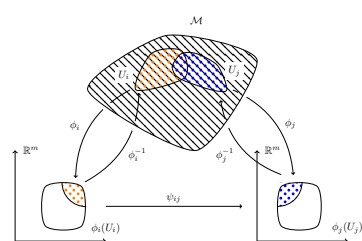

Figure 1: Schematic of a manifold $\mathcal{M}$ and open subsets $U_i$ and $U_j$. An open chart is a homeomorphism of an open subset of the manifold onto an open subset of the Euclidean hyperplane. Here, $\psi_{ij}$ is a transition function.

**Hausdorff and Gromov-Hausdorff Distances for Comparing Manifolds.** The Hausdorff distance between two subsets of a metric space refers to the greatest distance between any point on the first set and its closest point on the second set [Jungeblut et al., 2021]. Given a metric space $X$ with metric $d_X$, and two subsets $A$ and $B$, we can define the *Hausdorff distance* between $A$ and $B$ in $X$ by

$$d_{\mathrm{H}}^X(A, B) = \max \left( \sup_{a \in A} d_X(a, B), \sup_{b \in B} d_X(b, A) \right). \tag{1}$$

A priori this quantity may be infinite. Hence we will restrict to compact subsets $A$ and $B$. In this case, we can equivalently define $d_{\mathrm{H}}(A, B)$ as the smallest real number $c \geqslant 0$ such that for every $a \in A$ and every $b \in B$ there exist $a' \in A$ and $b' \in B$ such that both $d_X(a, b')$ and $d_X(a', b)$ are at most $c$.

We note that the previous definition does not require any differentiable structures on $X$, $A$ and $B$. They can be merely metric spaces. This generality allows us to distinguish the metric properties of Euclidean, hyperbolic and spherical geometries beyond analytic notions such as curvature. However, the definition in Equation 1 only allows to compare spaces $A$ and $B$ that are embedded in a certain metric space $X$. The notion of distance that we shall consider is the Gromov-Hausdorff distance, which we define below in Equation 3.

Given a metric space $X$ and two isometric embeddings $f : A \to X$ and $g : B \to X$, we define

$$d_{\mathrm{H}}^{X,f,g}(A, B) = d_{\mathrm{H}}^X(f(A), g(B)). \tag{2}$$

---

[2]For a high level description of the proposed framework, we recommend skipping to Section 3

Now, given two metric spaces $A$ and $B$, we denote by $\mathrm{ES}(A, B)$ (standing for "embedding spaces of $A$ and $B$") as the triple $(X, f, g)$ where $X$ is a metric space and $f : A \to X$ and $g : B \to X$ are isometric embeddings. We define the *Gromov-Hausdorff distance between $A$ and $B$* as:

$$\mathrm{d}_{\mathrm{GH}}(A, B) = \inf_{(X, f, g) \in \mathrm{ES}(A, B)} \mathrm{d}_{\mathrm{H}}^{X, f, g}(f(A), g(B)). \tag{3}$$

We should note that, since we assume that both $A$ and $B$ are compact, there is a trivial upper bound for their Gromov-Hausdorff distance in terms of their diameters. The *diameter* of a metric space $Y$ is defined to be $\mathrm{diam}(Y) = \sup_{y, y' \in Y} d_Y(y, y')$. Given $a_0 \in A$ and $b_0 \in B$, we can define the isometric embeddings $f : A \to A \times B$ and $g : B \to A \times B$ given by $f(a) = (a, b_0)$ and $g(b) = (a_0, b)$. It is easy to see that $\mathrm{d}_{\mathrm{H}}^{A \times B, f, g}(A, B) \leqslant \max(\mathrm{diam}(A), \mathrm{diam}(B))$. Since the triple $(A \times B, f, g)$ belongs to $\mathrm{ES}(A, B)$, we can estimate $\mathrm{d}_{\mathrm{GH}}(A, B) \leqslant \max(\mathrm{diam}(A), \mathrm{diam}(B))$. To compare $\mathbb{E}^n$, $\mathbb{H}^n$ and $\mathbb{S}^n$, we propose taking closed balls of radius one in each space. Since balls of radius one in any of these spaces are homogeneous Riemannian manifolds, they are isometric to each other. By estimating or providing an upper bound for their Gromov-Hausdorff distance, we can compare the spaces. This notion of distance between candidate latent geometries will later be used to generate a search space for our framework. With exactly an analogous argument as before, we can notice that given two compact balls of radius one $B$ and $B'$, of centres $x_0$ and $x_0'$, we can embed $B$ into $B \times B'$ by the mapping $f : b \mapsto (b, x_0')$. From here, it is obvious to see that $\mathrm{d}_{\mathrm{H}}^{B \times B', f, \mathrm{id}}(B, B \times B') = 1$. In particular, this gives us the bound $\mathrm{d}_{\mathrm{GH}}(B, B') \leqslant 1$. This is the estimation we will take as the Gromov-Hausdorff distances of product manifolds that simply differ in one coordinate (such as, say, $\mathbb{E}^2$ and $\mathbb{E}^2 \times \mathbb{H}^2$).

# 3 Neural Latent Geometry Search: Latent Product Manifold Inference

In this section, we leverage ideas discussed in Section 2 to introduce a principled way to find the optimal latent product manifold. First, we introduce the problem formulation of NLGS. Next, we outline the strategy used to compute the Gromov-Hausdorff distance between product manifolds, and we discuss how this notion of similarity can be used in practice to construct a graph search space of latent geometries. Lastly, we explain how the Gromov-Hausdorff-informed graph search space can be used to perform NLGS via BO.

## 3.1 Problem Formulation

The problem of NLGS can be formulated as follows. Given a search space $\mathfrak{G}$ denoting the set of all possible latent geometries, and the objective function $L_{T, A}(g)$ which evaluates the performance of a given geometry $g$ on a downstream task $T$ for a machine learning model architecture $A$, the objective is to find an optimal latent geometry $g^*$:

$$g^* = \arg\min_{g \in \mathfrak{G}} L_{T, A}(g). \tag{4}$$

In our case we model the latent space geometry using product manifolds. Hence we effectively restrict Equation 4 to finding the optimal product manifold signature:

$$n_{\mathcal{P}}^*, \{d_i\}_{i \in n_{\mathcal{P}}^*}^*, \{K_i\}_{i \in n_{\mathcal{P}}^*}^* = \arg\min_{n_{\mathcal{P}} \in \mathbb{Z}, d_i \in \mathbb{Z}, K_i \in \mathbb{R}} L_{T, A}(n_{\mathcal{P}}, \{d_i\}_{i \in n_{\mathcal{P}}}, \{K_i\}_{i \in n_{\mathcal{P}}}), \tag{5}$$

where $n_{\mathcal{P}}$ is the number of model spaces composing the product manifold $\mathcal{P}$, $\{d_i\}_{i \in n_{\mathcal{P}}}$ are the dimensions of each of the model spaces of constant curvature, and $\{K_i\}_{i \in n_{\mathcal{P}}}$ their respective curvatures. We further simplify the problem by setting $d_i = 2$, $\forall i$, and by restricting ourselves to $K_i \in \{-1, 0, 1\}$, in order to limit the size of the hypothesis space.

## 3.2 Quantifying the Difference Between Product Manifolds

**Motivation.** From a computational perspective, we can think of the Hausdorff distance as a measure of dissimilarity between two point sets, each representing a discretized version of the two underlying continuous manifolds we wish to compare. Taha and Hanbury [2015] proposed an efficient algorithm to compute the exact Hausdorff distance between two point sets with nearly-linear complexity leveraging early breaking and random sampling in place of scanning. However, the original algorithm assumes that both point sets live in the same space and have the same dimensionality, which is a

limiting requirement. Gromov-Hausdorff distances, as opposed to usual Hausdorff distances, allow us to measure the distance between two metric spaces that a priori are not embedded in a common bigger ambient space. However, this has the caveat that they are not computable. For instance, for our application we must calculate the distance between each pair of the following three spaces: $\mathbb{E}^n$, $\mathbb{S}^n$ and $\mathbb{H}^n$. However, $\mathbb{S}^n$ does not isometrically embed in $\mathbb{E}^n$ and hence in order to compare $\mathbb{E}^n$ and $\mathbb{S}^n$, we must work in a higher dimensional ambient space such as $\mathbb{E}^{n+1}$. In the case of $\mathbb{H}^n$, finding an embedding to a Euclidean space is more complicated and is described in Appendix B.2 ($\mathbb{H}^n$ will be embedded isometrically into $\mathbb{E}^{6n-6}$). However, in this process there will be choices made about which embeddings to consider (in particular, we cannot exactly compute the infimum that appears in the definition of Gromov-Hausdorff distance in Equation 3). Likewise, using the original algorithm by Taha and Hanbury [2015] it is not possible to compute the Hausdorff distance between product manifolds based on an unequal number of model spaces, for instance, there is no way of computing the distance between $\mathbb{E}^n$ and $\mathbb{E}^n \times \mathbb{H}^n$. In this section we give an upper bound for $d_{\mathrm{GH}}(\mathbb{E}^n, \mathbb{S}^n)$, and then describe an algorithm to give an upper bound for $d_{\mathrm{GH}}(\mathbb{E}^n, \mathbb{H}^n)$ and $d_{\mathrm{GH}}(\mathbb{S}^n, \mathbb{H}^n)$.

**Strategy.** The spaces $\mathbb{E}^n$ and $\mathbb{S}^n$ isometrically embed into $\mathbb{E}^{6n-6}$ in many ways. This may seem redundant because both spaces already embed in $\mathbb{E}^{n+1}$. However, the interest of considering this higher dimensional Euclidean space is that $\mathbb{H}^n$ will also isometrically embed into it. Crucially, this will provide a common underlying space in which to compute Hausdorff distances between our geometries $\mathbb{E}^n$, $\mathbb{S}^n$ and $\mathbb{H}^n$, which will lead to an estimation of their mutual Gromov-Hausdorff distance. The embedding of $\mathbb{H}^n$ into $\mathbb{E}^{6n-6}$ that we shall describe appears in Henke and Nettekoven [1987] and is made explicit in [Blanuša, 1955]. We also refer the reader to the exposition [Brander, 2003, Chapter 5], which puts this results in a broader context while also summarising related advances on the topic of isometrically embedding homogeneous spaces into higher dimensional ones. For $n = 2$, we name this embedding $F : \mathbb{H}^2 \to \mathbb{E}^6$ (Appendix B.2). For simplicity, in our experiments in Section 4, we will work with product manifolds generated based on constant curvature model spaces of dimension $n = 2$.

Now we can summarise our strategy to estimate $d_{\mathrm{GH}}(B_{\mathbb{E}^2}, B_{\mathbb{H}^2})$ as follows (for $d_{\mathrm{GH}}(B_{\mathbb{H}^2}, B_{\mathbb{S}^2})$ it will be entirely analogous). The first step consists of approximating our infinite smooth spaces by finite discrete ones. For this, we consider several collections of points $\{P_i\}_{i \in I}$ in $\mathbb{E}^2$ that are sufficiently well distributed. The exponential map can be applied to the collection of points $\exp : T_0\mathbb{H}^2 \cong \mathbb{R}^2 \to B_{\mathbb{H}^2}$ to get several collections of points $Q$ in $B_{\mathbb{H}^2}$ (again, well distributed by construction). In addition, we will consider several isometric embeddings $f_k : B_{\mathbb{E}^2} \to \mathbb{R}^6$. Hence, we take

$$d_{\mathrm{GH}}(B_{\mathbb{E}^2}, B_{\mathbb{H}^2}) \approx \min_{i,j,k} d_{\mathrm{H}}^{\mathbb{R}^6, f_k, F}(P_i, Q_j) = \min_{i,j,k} d_{\mathrm{H}}^{\mathbb{R}^6}(f_k(P_i), F(Q_j)). \tag{6}$$

In Appendix B, we gradually unravel the previous formula and give explicit examples of the involved elements. In particular, in Appendix B.1 we explain how to generate points in the balls of radius one, and in Appendix B.2 how to describe the isometric embedding $F : B_{\mathbb{H}^2} \to \mathbb{E}^6$. The results obtained for the Gromov-Hausdorff distances between product manifolds are used to generate the graph search space introduced in the next section.

### 3.3 The Gromov-Hausdorff-Informed Graph Search Space

**Gromov-Hausdorff Edge Weights.** In Section 2, we used Cartesian products of constant curvature model spaces to generate candidate latent geometries. We now turn our attention to constructing a search space to find the optimal latent geometry. To do so, we first consider all possible combinations of product manifolds based on a given number of model spaces, represented by $n_s$. Furthermore, we denote the total number of products (model spaces) used to form the product manifold $\mathcal{P}$ with $n_p$. Conceptually, $n_s$ is the number of model space *types* used, while $n_p$ refers to the overall number, or *quantity*, of model spaces that form the resulting product space. For instance, if only the Euclidean plane, the hyperboloid, and the hypersphere

Table 1: Estimated Gromov-Hausdorff distances (up to two decimal places) between model spaces and corresponding edge weights in the graph search space.

| Comparison Pair | $d_{\mathrm{GH}}(\cdot)$ | $w_{(\cdot)}$ |
|---|---|---|
| $(\mathbb{E}^2, \mathbb{S}^2)$ | 0.23 | 4.35 |
| $(\mathbb{E}^2, \mathbb{H}^2)$ | 0.77 | 1.30 |
| $(\mathbb{S}^2, \mathbb{H}^2)$ | 0.84 | 1.20 |

are taken into account, then $n_s = 3$. If all product mani-
fold combinations are considered, the number of elements in the search space increases to $\sum_{i=1}^{n_p} n_s^i$. However, we assume *commutativity* for latent product manifolds, implying that the output of a trained neural network with a latent geometry $\mathcal{M}_i \times \mathcal{M}_j$ should be the same as that with $\mathcal{M}_j \times \mathcal{M}_i$. We refer to this concept as the *symmetry of parameterization*, as neural networks can rearrange the weight matrices of their neurons to achieve optimal performance for two equivalent latent manifolds. This assumption reduces the search space from growing exponentially to $\mathcal{O}(n_p^2)$, assuming three constant curvature model spaces are considered (see Appendix B.5 for a more complete explanation).

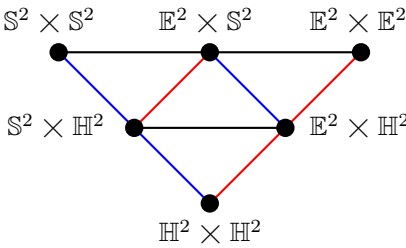

Figure 2: Slice of the graph search space for latent geometries of dimension 4: product manifolds obtained using 2 models spaces of dimension 2. The graph edges are shown in different colours to depict a different degree of connectivity (black: $w_{\mathbb{E}^2,\mathbb{S}^2}$, red: $w_{\mathbb{E}^2,\mathbb{H}^2}$, blue: $w_{\mathbb{S}^2,\mathbb{H}^2}$), this is determined by the inverse of the Gromov-Hausdorff distance between the different product manifolds.

We model the search space as a graph over which we perform BO. More formally, we consider a graph $\mathcal{G}$ given by $(\mathcal{V}, \mathcal{E})$, where $\mathcal{V} = \{v_i\}_{i=1}^N$ are the nodes of the graph and $\mathcal{E} = \{e_j\}_{j=1}^M$ is the set of edges, where edge $e_j = (v_i, v_j)$ connects nodes $v_i$ and $v_j$. The topology of the graph is encoded in its *adjacency matrix*, which we denote with $\mathbf{A} \in \mathbb{R}^{N \times N}$. In our case, we focus on a *weighted* and *undirected* graph, meaning that $\mathbf{A}_{i,j} = \mathbf{A}_{j,i}$, and $\mathbf{A}_{i,j}$ can take any positive real values. We consider a setting in which the function $\mathfrak{f}(\cdot)$ to minimize is defined over the nodes of the graph, and the objective of using BO is to find the node associated to the minimum value $v^* = \arg\min_{v \in \mathcal{V}} \mathfrak{f}(v)$.

In our setting, each node in the graph represents a different latent space product manifold, and the value at that node is the validation set performance of a neural network architecture using this latent geometry. We use the inverse of the Gromov-Hausdorff distance between product manifolds to obtain edge weights.

We denote by $w_{\mathcal{M}_1,\mathcal{M}_2}$ (edge weights) the inverse Gromov-Hausdorff distance between model spaces $\mathcal{M}_1$ and $\mathcal{M}_2$. The approximate values of these coefficients are presented in Table 1. In particular, the Gromov-Hausdorff distance $d_{\mathrm{GH}}(\mathbb{E}^2, \mathbb{S}^2)$ used to compute $w_{\mathbb{E}^2,\mathbb{S}^2}$ is derived analytically in Appendix B.3, while the remaining coefficients are obtained through computational approximations of Equation 6, which are depicted in Appendix B.4. Further, the Gromov-Hausdorff distance between manifolds of different dimensions is one (see Section 2).

In order to impose additional structure on the search space, we only allow connections between nodes corresponding to product manifolds which differ by a single model space. For example, within the same dimension, $\mathbb{S}^2 \times \mathbb{H}^2$ and $\mathbb{E}^2 \times \mathbb{H}^2$ would be connected with edge weighting $w_{\mathbb{E}^2,\mathbb{S}^2}$ while $\mathbb{S}^2 \times \mathbb{H}^2$ and $\mathbb{E}^2 \times \mathbb{E}^2$ would have no connection in the graph. Furthermore, product manifolds in different dimensions follow the same rule. For instance, we would have a connection of strength one between $\mathbb{E}^2 \times \mathbb{H}^2$ and $\mathbb{E}^2 \times \mathbb{H}^2 \times \mathbb{H}^2$, but no connection between, for instance, $\mathbb{E}^2$ and $\mathbb{E}^2 \times \mathbb{H}^2 \times \mathbb{H}^2$ or $\mathbb{S}^2$ and $\mathbb{E}^2 \times \mathbb{H}^2$. This construction induces a sense of directionality into the graph and generates clusters of product manifolds of the same dimension. Finally, it should be mentioned that in practice there are only four edge weights. For example, the connectivity strength between $\mathbb{E}^2 \times \mathbb{H}^2$ and $\mathbb{H}^2 \times \mathbb{H}^2$ is $w_{\mathbb{E}^2,\mathbb{H}^2}$ since $d_{\mathrm{GH}}(\mathbb{E}^2 \times \mathbb{H}^2, \mathbb{H}^2 \times \mathbb{H}^2) = d_{\mathrm{GH}}(\mathbb{E}^2, \mathbb{H}^2)$ given that $d_{\mathrm{GH}}(\mathbb{H}^2, \mathbb{H}^2) = 0$. Visual representations of the graph search space can be found Figures 2 and 3. Note that both figures use the same colour scheme, and in Figure 3 there is an increase in the dimen-

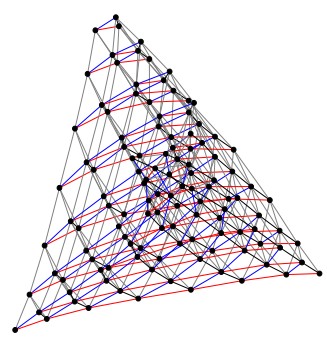

Figure 3: Example graph search space for product manifolds composed of up to seven model spaces. Manifolds of different dimensionality are connected with edges coloured grey. Node labels have been omitted for visual clarity.

sionality of the product manifolds from left to right
(e.g. on the top left corner, one can see a triangle
corresponding to the three model spaces). We refer the reader to Appendix B.6 for additional
visualizations.

**Bayesian Optimization over the Graph Search Space.**   We aim to find the minimum point within
the graph search space through the use of BO (see Appendix C). While performing BO on graphs
allows us to search for the minimum over a categorical search space, some of the key notions used
in BO over Euclidian spaces do not translate directly when operating over a graph. For example,
notions of similarity or "closeness" which are trivially found in Euclidian space through the $\ell_1$ or $\ell_2$
norms require more careful consideration when using graphs.

In our setting, we employ a *diffusion kernel* [Smola and Kondor, 2003, Kondor and Vert, 2004]
to compute the similarity between the nodes in the graph. The diffusion kernel is based on the
eigendecomposition of the graph Laplacian $\mathbf{L} \in \mathbb{R}^{N \times N}$, defined as $\mathbf{L} = \mathbf{D} - \mathbf{A}$ where $\mathbf{D}$ is the degree
matrix of the graph. In particular, the eigendecomposition of the Laplacian is given by $\mathbf{L} = \mathbf{U}\mathbf{\Lambda}\mathbf{U}^{\mathrm{T}}$,
where $\mathbf{U} = [\mathbf{u}_1, \dots, \mathbf{u}_N]$ is a matrix containing eigenvectors as columns and $\mathbf{\Lambda} = \mathrm{diag}(\lambda_1, \dots, \lambda_N)$
is a diagonal matrix containing increasingly ordered eigenvalues. The *covariance matrix* used to
define the GP over the graph is given by

$$\mathcal{K}(\mathcal{V}, \mathcal{V}) = \mathbf{U}e^{-\beta\mathbf{\Lambda}}\mathbf{U}^{\mathrm{T}}, \tag{7}$$

where $\beta$ is the *lengthscale* parameter. Our approach is therefore conceptually similar to that of [Oh
et al., 2019], which employ a diffusion kernel to carry out a NAS procedure on a graph Cartesian
product. We highlight, however, that the main contribution of this work is the construction of the
graph search space, and we employ this search procedure to showcase the suitability of our method
in finding the optimal latent product manifold. For reference, we note that other works employing
Bayesian optimization in graph-related settings include Cui and Yang [2018], Como et al. [2020],
Ma et al. [2019], Ru et al. [2020], Cui et al. [2020], Wan et al. [2021].

## 4   Experimental Setup and Results

In this section, a detailed outline of the experimental results is provided. It comprises of experiments
performed on synthetic datasets, as well as experimental validation on custom-designed real-world
datasets obtained by morphing the latent space of mixed-curvature autoencoders and latent graph
inference. The results demonstrate that the Gromov-Hausdorff-informed graph search space can be
leveraged to perform NLGS across a variety of tasks and datasets.

### 4.1   Synthetic Experiments on Product Manifold Inference

In order to evaluate the effectiveness of the proposed graph search space, we conduct a series of tests
using synthetic data. We wish to present a setting for which the latent optimal product manifold
is known by construction. To do so, we start by generating a random vector $\mathbf{x}$ and mapping it to a
"ground truth" product manifold $\mathcal{P}_T$, which we choose arbitrarily, using the corresponding exponential
map. The resulting projected vector is then decoded using a neural network with frozen weights,
$f_\theta$, to obtain a reference signal $y^{\mathcal{P}_T}$, which we wish to recover using NLGS. To generate the rest
of the dataset, we then consider the same random vector but map it to a number of other product
manifolds $\{\mathcal{P}_i\}_{i \in n_{\mathcal{P}}}$ with the same number of model spaces as $\mathcal{P}_T$ but different signatures. Decoding
the projected vector through the same neural network yields a set of signal, $\{y^{\mathcal{P}_i}\}_{i \in n_{\mathcal{P}}}$. We use the
aforementioned signals, to set the value associated with each node of the graph search space to be
$MSE(y^{\mathcal{P}_T}, y^{\mathcal{P}_i})$. In this way, our search algorithm should aim to find the node in the graph that
minimizes the error and hence find the latent geometry that recovers the original reference signal.

Our method consists in using BO over our Gromov-Hausdorff-informed search space. We compare it
against random search, and what we call "Naive BO", which performs BO over a fully-connected
graph which disregards the Gromov-Hausdorff distance between candidate latent geometries. The
figures presented, namely Figures 4 and 5, display the results. Each plot illustrates the performance
of the algorithms and baselines as they select different optimal latent product manifolds denoted as
$\mathcal{P}_T$. Notably, the figures demonstrate that the algorithm utilizing the Gromov-Hausdorff-informed
search space surpasses all other baselines under consideration and consistently achieves the true

function minimum. Figure 4 generally requires fewer iterations compared to Figure 5 to attain the global minimum, primarily due to the smaller size of the search space. It is important to note that our benchmark solely involves the same algorithm but with a change in the search space to a topologically uninformative one. This modification allows us to evaluate the impact of incorporating this information into the optimization process. We have not considered other benchmarks such as evolutionary algorithms since they always rely on a notion of distance between sampled points. Furthermore, as there are no existing algorithms to compute the distance between arbitrary product manifolds, we have not included these methods in our benchmark as they would simply converge to random search. All results are shown on a log scale, and we apply a small offset of $\varepsilon = 10^{-3}$ to the plots to avoid computing $\log(0)$ when the reference signal is found.

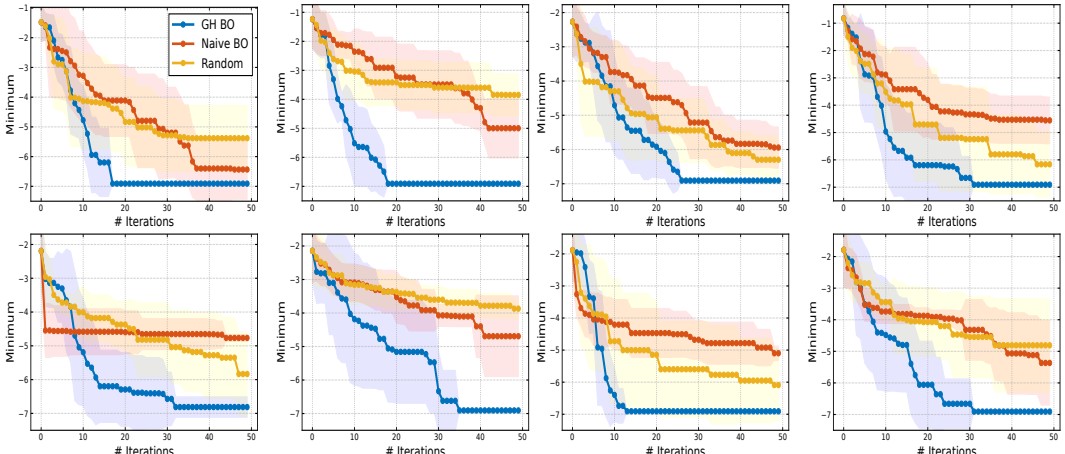

Figure 4: Results (mean and standard deviation over 10 runs) for candidate latent geometries involving product manifolds composed of 13 model spaces. For each plot a different ground truth product manifold $\mathcal{P}_T$ is used to generate the reference signal.

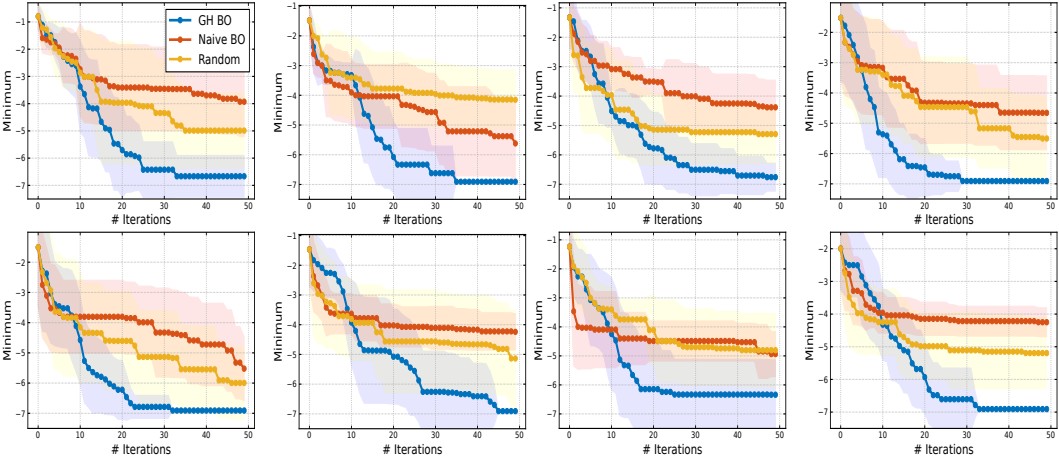

Figure 5: Results (mean and standard deviation over 10 runs) for candidate latent geometries involving product manifolds composed of 20 model spaces. Same setup as above.

## 4.2 Experiments on Real-World Datasets

To further validate our method, we conduct additional tests using image and graph datasets. Specifically, we focus on image reconstruction and node classification tasks.

### Image Reconstruction with Autoencoder

We consider four well-known image datasets: MNIST [Deng, 2012], CIFAR-10 [Krizhevsky et al., 2014], Fashion MNIST [Xiao et al., 2017] and eMNIST [Cohen et al., 2017]. Some of these datasets have been shown to benefit from additional topological priors, see Khrulkov et al. [2020] and Moor et al. [2020]. We use an autoencoder, detailed more thoroughly in Appendix D, to encode the image in a low dimensional latent space and then reconstruct it based on its latent representation. To test the performance of different latent geometries, we project the latent vector onto the product manifold being evaluated and use the reconstruction loss at the end of training as the reference signal to use in the graph search space. We consider a search space size of $n_p = 7$ for MNIST and $n_p = 8$ for CIFAR-10, Fashion MNIST and eMNIST. The results are displayed in Figure 6. In line with previous experiments, the Gromov-Hausdorff-informed search graph enables us to find better solutions in a smaller amount of evaluations.

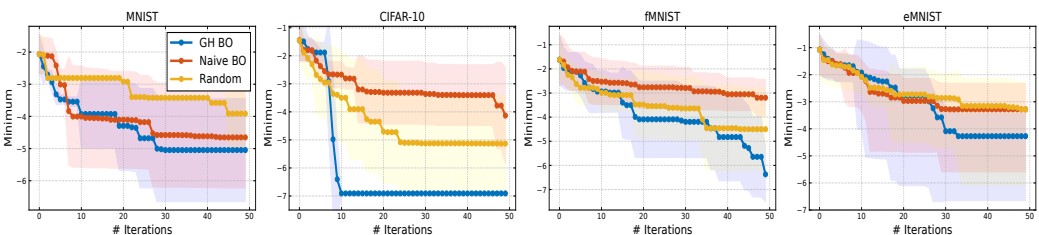

Figure 6: Results (mean and standard deviation over 10 runs) for image reconstruction tasks.

### Latent Graph Inference

Finally, we consider searching for the optimal product manifold for a graph neural network node classification task using latent graph inference. In particular, we build upon previous work by Sáez de Ocáriz Borde et al. [2023b] which used product manifolds to produce richer embeddings spaces for latent graph inference. We consider the Cora [Sen et al., 2008] and Citeseer [Giles et al., 1998] citation network datasets, and a search space consisting of product manifolds of up to seven model spaces $n_p = 7$. The aim is to find the latent space which gives the best results for the node classification problem using minimal query evaluations. Performing BO alongside the Gromov-Hausdorff informed search space gives a clear competitive advantage over Naive BO and random search in the case of Cora. For Citeseer, the experiments do not give such a clear-cut improvement, which can be attributed to the lack of smoothness of the signal on the graph search space and its incompatibility with the intrinsic limitations of the diffusion kernel.

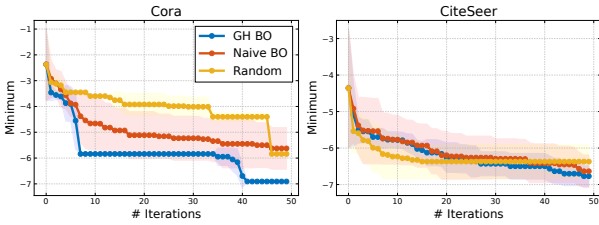

Figure 7: Results (mean and standard deviation over 10 runs) for latent graph inference datasets.

# 5 Conclusion

In this work, we have introduced *neural latent geometry search (NLGS)*, a novel problem formulation that consists in finding the optimal latent space geometry of machine learning algorithms using minimal query evaluations. In particular, we have modeled the latent space using product manifolds based on Cartesian products of constant curvature model spaces. To find the optimal product manifold, we propose using Bayesian Optimization over a graph search space. The graph is constructed based on a principled measure of similarity, utilizing the Gromov-Hausdorff distance from metric geometry. The effectiveness of the proposed method is demonstrated through our experiments conducted on a variety of tasks, based on custom-designed synthetic and real-world datasets.

**Limitations and Future Work.** While the NLGS framework is general, we have restricted ourselves to using product manifolds of constant curvature model spaces to model the geometry of the latent space. Furthermore, we have only considered curvatures $\{-1, 0, 1\}$, and model spaces of dimension two in order for the optimization problem to be tractable. In future research, there is potential to explore alternative approaches for modeling the latent space manifold. Additionally, the field of Bayesian Optimization over graphs is still in its early stages. Incorporating recent advancements in the area of kernels on graphs [Borovitskiy et al., 2021, Zhi et al., 2023] could lead to improved performance. In our current research, our emphasis is on introducing NLGS and providing an initial solution under a set of simplifying assumptions related to the potential latent manifolds available and the optimization algorithm. The investigation of the impact of using different similarity measures to compare latent structures also remains a subject for future research.

## Societal Impact Statement

This work is unlikely to result in any harmful societal repercussions. Its primary potential lies in its ability to enhance and advance existing data modelling and machine learning methods.

## Acknowledgement

AA thanks the Rafael del Pino Foundation for financial support. AA and HSOB thank the Oxford-Man Institute for computational resources. AA acknowledges the G-Research grant for travel assistance. XD acknowledges support from the Oxford-Man Institute and the EPSRC (EP/T023333/1).

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

# A   Additional Background

## A.1   Differential Geometry, Riemmanian Manifolds, and Product Manifolds

Differential geometry is a mathematical discipline that employs rigorous mathematical techniques, including calculus and other mathematical tools, to investigate the properties and structure of smooth manifolds. These manifolds, despite bearing a local resemblance to Euclidean space, may exhibit global topological features that distinguish them from Euclidean space. The fundamental objective of differential geometry is to elucidate the geometric properties of these spaces, such as curvature, volume, and length, through rigorous mathematical analysis and inquiry.

A Riemannian manifold generalizes Euclidean space to more curved spaces, and it plays a critical role in differential geometry and general relativity. A Riemannian manifold is a smooth manifold $M$ equipped with a Riemannian metric $g$, which assigns to each point $p$ in $M$ an inner product on the tangent space $T_pM$. This inner product is a smoothly varying positive definite bilinear form on the tangent space given by $g_p : T_pM \times T_pM \to \mathbb{R}$, which satisfies the following properties: symmetry, $g_p(X, Y) = g_p(Y, X)$ for all $X, Y \in T_pM$; linearity, $g_p(aX + bY, Z) = ag_p(X, Z) + bg_p(Y, Z)$ for all $X, Y, Z \in T_pM$ and scalars $a, b \in \mathbb{R}$; positive definiteness: $g_p(X, X) > 0$ for all non-zero $X \in T_pM$. The metric $g_p$ induces a norm $||.||_p$ on $T_pM$ given by $||X||_p = \sqrt{g_p(X, X)}$. This norm allows us to define the length of curves on the manifold, and hence a notion of distance between points. Specifically, given a curve $c : [0, 1] \to M$ and a partition $0 = t_0 < t_1 < ... < t_n = 1$, the length of the curve $c$ over the partition is given by: $L(c) = \sum ||c'(t_i)||c(t_i)$, where $c'(t_i)$ denotes the tangent vector to $c$ at time $t_i$, and $|| \cdot ||c(t_i)$ denotes the norm induced by the metric at the point $c(t_i)$. The total length of the curve is obtained by taking the limit of the sum as the partition becomes finer.

On the other hand, a product manifold results from taking the Cartesian product of two or more manifolds, resulting in a manifold with a natural product structure. This enables researchers to examine the local and global geometry of the product manifold by studying the geometry of its individual factors. Riemannian manifolds and product manifolds are essential mathematical concepts that have far-reaching applications in diverse fields such as physics, engineering, and computer science. In the appendix of this paper, we provide a more in-depth look at these concepts, including their properties and significance.

## A.2   Constant Curvature Model Spaces

A manifold $\mathcal{M}$ is a topological space that can be locally identified with Euclidean space using smooth maps. In Riemannian geometry, a Riemannian manifold or Riemannian space $(\mathcal{M}, g)$ is a real and differentiable manifold $\mathcal{M}$ where each tangent space has an associated inner product $g$ known as the Riemannian metric. This metric varies smoothly from point to point on the manifold and allows for the definition of geometric properties such as lengths of curves, curvature, and angles. The Riemannian manifold $\mathcal{M} \subseteq \mathbb{R}^N$ is a collection of real vectors that is locally similar to a linear space and exists within the larger ambient space $\mathbb{R}^N$.

Curvature is effectively a measure of geodesic dispersion. When there is no curvature geodesics stay parallel, with negative curvature they diverge, and with positive curvature they converge. Constant curvature model spaces are Riemannian manifolds with a constant sectional curvature, which means that the curvature is constant in all directions in a 2D plane (this can be generalized to higher dimensions). These spaces include the Euclidean space, the hyperbolic space, and the sphere. The Euclidean space has zero curvature, while the hyperbolic space has negative curvature and the sphere has positive curvature. Constant curvature model spaces are often used as reference spaces for comparison in geometric analysis and machine learning on non-Euclidean data.

Euclidean space,

$$\mathbb{E}^{d_{\mathbb{E}}}_{K_{\mathbb{E}}} = \mathbb{R}^{d_{\mathbb{E}}} \tag{8}$$

is a flat space with curvature $K_{\mathbb{E}} = 0$. We note that in this context, the notation $d_{\mathbb{E}}$ is used to represent the dimensionality. In contrast, hyperbolic and spherical spaces possess negative and positive curvature, respectively. We define hyperboloids $\mathbb{H}^{d_{\mathbb{H}}}_{K_{\mathbb{H}}}$ as

$$\mathbb{H}_{K_{\mathbb{H}}}^{d_{\mathbb{H}}} = \{\mathbf{x}_p \in \mathbb{R}^{d_{\mathbb{H}}+1} : \langle \mathbf{x}_p, \mathbf{x}_p \rangle_{\mathcal{L}} = 1/K_{\mathbb{H}}\}, \tag{9}$$

where $K_{\mathbb{H}} < 0$ and $\langle \cdot, \cdot \rangle_{\mathcal{L}}$ is the Lorentz inner product

$$\langle \mathbf{x}, \mathbf{y} \rangle_{\mathcal{L}} = -x_1 y_1 + \sum_{j=2}^{d_{\mathbb{H}}+1} x_j y_j, \quad \forall \mathbf{x}, \mathbf{y} \in \mathbb{R}^{d_{\mathbb{H}}+1}, \tag{10}$$

and hyperspheres $\mathbb{S}_{K_{\mathbb{S}}}^{d_{\mathbb{S}}}$ as

$$\mathbb{S}_{K_{\mathbb{S}}}^{d_{\mathbb{S}}} = \{\mathbf{x}_p \in \mathbb{R}^{d_{\mathbb{S}}+1} : \langle \mathbf{x}_p, \mathbf{x}_p \rangle_2 = 1/K_{\mathbb{S}}\}, \tag{11}$$

where $K_{\mathbb{S}} > 0$ and $\langle \cdot, \cdot \rangle_2$ is the standard Euclidean inner product

$$\langle \mathbf{x}, \mathbf{y} \rangle_2 = \sum_{j=1}^{d_{\mathbb{S}}+1} x_j y_j, \quad \forall \mathbf{x}, \mathbf{y} \in \mathbb{R}^{d_{\mathbb{S}}+1}, \tag{12}$$

Table 2 summarizes the key operators in Euclidean, hyperbolic, and spherical spaces with arbitrary curvatures. It is worth noting that the three model spaces discussed above can cover any curvature value within the range of $(-\infty, \infty)$. However, there is a potential issue with the hyperboloid and hypersphere becoming divergent as their respective curvatures approach zero. This results in the manifolds becoming flat, and their distance and metric tensors do not become Euclidean at zero-curvature points, which could hinder curvature learning. Therefore, stereographic projections are considered to be suitable alternatives to these spaces as they maintain a non-Euclidean structure and inherit many of the properties of the hyperboloid and hypersphere.

Table 2: Relevant operators (exponential maps and distances between two points) in Euclidean, hyperbolic, and spherical spaces with arbitrary constant curvatures.

| Space Model | $exp_{\mathbf{x}_p}(\mathbf{x})$ | $\mathfrak{d}(\mathbf{x}, \mathbf{y})$ |
|---|---|---|
| $\mathbb{E}$, Euclidean | $\mathbf{x}_p + \mathbf{x}$ | $\|\mathbf{x} - \mathbf{y}\|_2$ |
| $\mathbb{H}$, hyperboloid | $\cosh\left(\sqrt{-K_{\mathbb{H}}}\|\mathbf{x}\|\right)\mathbf{x}_p + \sinh\left(\sqrt{-K_{\mathbb{H}}}\|\mathbf{x}\|\right)\frac{\mathbf{x}}{\sqrt{-K_{\mathbb{H}}}\|\mathbf{x}\|}$ | $\frac{1}{\sqrt{-K_{\mathbb{H}}}}\text{arccosh}\left(K_{\mathbb{H}}\langle \mathbf{x}, \mathbf{y} \rangle_{\mathcal{L}}\right)$ |
| $\mathbb{S}$, hypersphere | $\cos\left(\sqrt{K_{\mathbb{S}}}\|\mathbf{x}\|\right)\mathbf{x}_p + \sin\left(\sqrt{K_{\mathbb{S}}}\|\mathbf{x}\|\right)\frac{\mathbf{x}}{\sqrt{K_{\mathbb{S}}}\|\mathbf{x}\|}$ | $\frac{1}{\sqrt{K_{\mathbb{S}}}}\arccos\left(K_{\mathbb{S}}\langle \mathbf{x}, \mathbf{y} \rangle_2\right)$ |

### A.3 Constructing Product Manifolds

In this appendix, we provide additional details about the generation of product manifolds. In mathematics, a product manifold is a type of manifold obtained by taking the Cartesian product of two or more manifolds. Informally, a product manifold can be thought of as a space formed by taking multiple smaller spaces and merging them in a way that maintains their individual structures. For instance, the surface of a sphere can be seen as a product manifold consisting of two real lines that intersect at each point on the sphere. Product manifolds play a crucial role in various fields of mathematics, including physics, topology, and geometry.

A product manifold can be defined as the Cartesian product:

$$\mathcal{P} = \underset{i=1}{\overset{n_{\mathcal{P}}}{\LARGE\times}} \mathcal{M}_{K_i}^{d_i} \tag{13}$$

Here, $K_i$ and $d_i$ represent the curvature and dimensionality of the space, respectively. Points $\mathbf{x}_p \in \mathcal{P}$ are expressed using their coordinates:

$$\mathbf{x}_p = concat\left(\mathbf{x}_p^{(1)}, \mathbf{x}_p^{(2)}, ..., \mathbf{x}_p^{(n_\mathcal{P})}\right) : \mathbf{x}_p^{(i)} \in \mathcal{M}_{K_i}^{d_i}. \tag{14}$$

Also, the metric of the product manifold decomposes into the sum of the constituent metrics

$$g_\mathcal{P} = \sum_{i=1}^{n_\mathcal{P}} g_i, \tag{15}$$

hence, $(\mathcal{P}, g_\mathcal{P})$ is also a Riemannian manifold. It should be noted that the signature of the product space is parametrized with several degrees of freedom. These include the number of components used in the product space, the type of model spaces employed, their dimensionality, and curvature. If we restrict $\mathcal{P}$ to be composed of the Euclidean plane $\mathbb{E}_{K_\mathbb{E}}^{d_\mathbb{E}}$, hyperboloids $\mathbb{H}_{K_j^\mathbb{H}}^{d_j^\mathbb{H}}$, and hyperspheres $\mathbb{S}_{K_k^\mathbb{S}}^{d_k^\mathbb{S}}$ of constant curvature, we can rewrite Equation 13 as

$$\mathcal{P} = \mathbb{E}_{K_\mathbb{E}}^{d_\mathbb{E}} \times \left(\underset{j=1}{\overset{n_\mathbb{H}}{\times}} \mathbb{H}_{K_j^\mathbb{H}}^{d_j^\mathbb{H}}\right) \times \left(\underset{k=1}{\overset{n_\mathbb{S}}{\times}} \mathbb{S}_{K_k^\mathbb{S}}^{d_k^\mathbb{S}}\right), \tag{16}$$

where $K_\mathbb{E} = 0$, $K_j^\mathbb{H} < 0$, and $K_k^\mathbb{S} > 0$. Hence, $\mathcal{P}$ would have a total of $1 + n_\mathbb{H} + n_\mathbb{S}$ component spaces, and total dimension $d_\mathbb{E} + \Sigma_{j=1}^{n_\mathbb{H}} d_j^\mathbb{H} + \Sigma_{k=1}^{n_\mathbb{S}} d_k^\mathbb{S}$. Distances between two points in the product manifold can be computed aggregating the distance contributions $\mathfrak{d}_i$ from each manifold composing the product manifold:

$$\mathfrak{d}_\mathcal{P}(\overline{\mathbf{x}}_{p_1}, \overline{\mathbf{x}}_{p_2}) = \sqrt{\sum_{i=1}^{n_\mathcal{P}} \mathfrak{d}_i\left(\overline{\mathbf{x}}_{p_1}^{(i)}, \overline{\mathbf{x}}_{p_2}^{(i)}\right)^2}. \tag{17}$$

The overline denotes that the points $\mathbf{x}_{p_1}$ and $\mathbf{x}_{p_2}$ have been adequately projected on to the product manifold using the exponential map before computing the distances.

### A.4 The Rationale Behind using Product Manifolds of Model Spaces

Most arbitrary Riemannian manifolds do not have closed-form solutions for their exponential maps and geodesic distances between points, as well as for other relevant mathematical notions. The exponential map takes a point on the manifold and exponentiates a tangent vector at that point to produce another point on the manifold. The geodesic distance between two points on a manifold is the shortest path along the manifold connecting those points.

For simple, well-studied manifolds like Euclidean space, hyperbolic space, and spherical space, closed-form solutions for exponential maps and geodesic distances are available. This is because these manifolds have constant curvature, which allows for more straightforward calculations. However, for most other manifolds, the exponential map and geodesic distances must typically be computed numerically or approximated using specialized algorithms. These computations often involve solving differential equations, and depending on the manifold's curvature and geometry, these solutions can be quite complex and may not have closed-form expressions. For certain types of manifolds, like product manifolds where each factor is a well-understood manifold, the computation of exponential maps and geodesic distances can sometimes be simplified due to the separability of the metric. However, scaling these methods to more general, complex manifolds can be challenging and computationally intensive.

As mentioned, closed-form solutions for exponential maps and geodesic distances are not common for arbitrary Riemannian manifolds, and computational methods often rely on numerical techniques and algorithms due to the complexity of these calculations. These properties have made product manifolds the most attractive tool to endow the latent space with richer structure while retaining computational traceability, see [Gu et al., 2018, Skopek et al., 2019, Sáez de Ocáriz Borde et al., 2023b,a, Zhang et al., 2020, Fumero et al., 2021, Pfau et al., 2020]. This motivates us to use product manifolds in our setup, which keeps our work in line with the state-of-the-art approaches in the literature. We highlight that one of the central contributions of our paper is to shed light on the problem of NLGS, and provide an initial solution to the problem in the context of the current state-of-the-art in geometric

latent space modelling. Future research avenues could explore generalizations of this which account for new techniques that emerge in the literature, potentially comparing a more general set of latent manifolds.

### A.5   Geometry of Negative Curvature: Further Discussion

Lastly, in this appendix, we aim to provide an intuition regarding closed balls in spaces of curvature smaller than zero. The hyperbolic spaces we consider in this article are hyperbolic manifolds (that is, hyperbolic in the differentiable sense). However, there are other generalisations of the notion of hyperbolicity that carry over to the setting of geodesic metric spaces (such as Gromov hyperbolicity). This theory unifies the two extremes of spaces of negative curvature, the real hyperbolic spaces $\mathbb{H}^n$ and trees. Recall that a tree is a simple graph that does not have closed paths. Trees provide a great intuition for the metric phenomena that we should expect in spaces of negative curvature. For example, let us denote by $G$ an infinite planar grid (observe that this graph is far from being a tree), and let us denote by $T$ a regular tree of valency 4 (this means that every vertex in $T$ is connected to exactly other four vertices). The 2-dimensional Euclidean space $\mathbb{E}^2$ is quasi-isometric to $G$ and the 2-dimensional real hyperbolic space $\mathbb{H}^2$ is quasi-isometric (but not isometric) to $T$. This observation is due to Švarc and Milnor. Recall that the graphs $G$ and $T$ are endowed with the natural path metric. A ball of radius $n > 0$ in $G$ has between $n^2$ and $4n^2$ points (polynomial in $n$), while a ball of radius $n$ in $T$ has exactly $(4^n - 1)/3$ points (exponential in $n$). Now let us come back to $\mathbb{H}^3$, where the "continuous" analogue of this principle is also true; that is, that there exists a constant $c > 1$ such that for every $n > 0$, one needs at least $c^n$ balls of radius one to cover a ball of radius $n$ in $\mathbb{H}^3$. If we only want to understand the local geometry of $\mathbb{H}^3$, we can re-scale this principle to say the following. Let $B_1^{\mathbb{H}^3}$ denote the ball of radius one in $\mathbb{H}^3$. For all positive integers $n > 0$, one needs at least $c^n$ balls of radius $1/n$ to cover $B_1^{\mathbb{H}_3}$. This tree-like behaviour should be compared with the opposite principle that holds in $\mathbb{E}^3$, being that for all $n > 0$, one needs at most $8n^3$ (a polynomial bound) of balls of radius one to cover $B_1^{\mathbb{E}^3}$. Essentially, we should think of $B_1^{\mathbb{H}_3}$ as a manifold that is isomorphic to $B_1^{\mathbb{E}^3}$ which is difficult to cover by balls of smaller radius, in fact, it will be coarsely approximated by a finite tree of balls of small radius (the underlying tree being bigger as the chosen radius is smaller). Furthermore, the philosophy that travelling around $B_1^{\mathbb{H}_3}$ becomes more and more expensive as we leave the origin is made accurate with the consideration of divergence functions: geodesics in $\mathbb{H}^3$ witness exponential divergence. More precisely, there exists a constant $c > 1$ such that for every $r > 0$ and for every geodesic $\gamma : \mathbb{R} \longrightarrow \mathbb{H}^3$ passing through the origin at time $t = 0$ it is true that if $\alpha$ is a path that connects $\gamma(-r)$ and $\gamma(r)$ and that lies entirely outside of the ball of radius $d(0, \gamma(r))$, then $\alpha$ must have length at least $c^r$. Again, this is better appreciated when put in comparison with the Euclidean space $\mathbb{E}^3$. Any geodesic (any line) $\gamma : \mathbb{R} \longrightarrow \mathbb{E}^3$ passing through the origin at time $t = 0$ will have the property that for all $r > 0$ there is a path of length $\pi r + \varepsilon$, with $\varepsilon > 0$ arbitrarily small, that lies entirely outside of the ball of radius $r$ with centre at the origin. The length $\pi r + \varepsilon$ is linear in $r$, as opposed to the exponential length that is required in hyperbolic spaces.

## B   Further Details on the Gromov-Hausdorff Algorithm for Neural Latent Geometry Search

As discussed in Section 3, there are limitations to the Hausdorff distance as a metric for comparing point sets that represent discretized versions of continuous manifolds. The original algorithm by Taha and Hanbury [2015] for computing the Hausdorff distance assumes that the point sets reside in the same space with the same dimensionality, which is a limiting requirement. However, the Gromov-Hausdorff distance proposed in this work allows us to compare metric spaces that are not embedded in a common ambient space, but it is not exactly computable. We will focus on providing an upper bound for $d_{\mathrm{GH}}(\mathbb{E}^n, \mathbb{S}^n)$ and describing an algorithm for obtaining upper bounds for $d_{\mathrm{GH}}(\mathbb{E}^n, \mathbb{H}^n)$ and $d_{\mathrm{GH}}(\mathbb{S}^n, \mathbb{H}^n)$ too. As previously mentioned, $\mathbb{E}^n$ and $\mathbb{S}^n$ can be isometrically embedded into $\mathbb{E}^{6n-6}$ in many ways, which provides a common underlying space for computing Hausdorff distances between $\mathbb{E}^n$, $\mathbb{S}^n$, and $\mathbb{H}^n$. The embeddings of different spaces require making choices: in particular, there is no way to exactly compute the infimum in the definition of Gromov-Hausdorff distance given in Section 2:

$$d_{\mathrm{GH}}(A, B) = \inf_{(X,f,g)\in\mathrm{ES}(A,B)} d_{\mathrm{H}}^{X,f,g}(f(A), g(B)). \tag{18}$$

so we resort to numerical approximations which are later described in this appendix.

To estimate the mutual Gromov-Hausdorff distance between the infinite smooth spaces $\mathbb{E}^2$ and $\mathbb{H}^2$, the first step is to approximate them using finite discrete versions. This is done by considering well-distributed collections of points in $\mathbb{E}^2$ and $\mathbb{H}^2$, obtained by applying the exponential map to a collection of points in $\mathbb{R}^2$. Multiple isometric embeddings of $\mathbb{E}^2$ into $\mathbb{R}^6$ are also considered. The estimation is obtained by computing the minimum of the Hausdorff distance between the point sets obtained from the collections in $\mathbb{R}^6$ and the isometric embedding of $\mathbb{H}^2$ into $\mathbb{R}^6$. The same applies for spherical space.

## B.1 Generating Points in the Corresponding Balls of Radius One

All our computations will be done in dimension two, so we will simply precise how to generate points in $B_{\mathbb{E}^2}$, $B_{\mathbb{S}^2}$ and $B_{\mathbb{H}^2}$. For $B_{\mathbb{E}^2}$ and $B_{\mathbb{S}^2}$, we will use very elementary trigonometry. For $B_{\mathbb{H}^2}$, we will need an explicit description of the exponential map of $\mathbb{H}^2$. Using the descriptions

$$B_{\mathbb{E}^2} = \{(r\cos(t), r\sin(t)) : r \in [0, 1], t \in [0, 2\pi]\}, \tag{19}$$

and

$$B_{\mathbb{S}^2} = \{(\sin(\beta)\cos(\alpha), \sin(\beta)\sin(\alpha), \cos(\beta)) : \alpha \in [0, 2\pi), \beta \in [0, 1]\}, \tag{20}$$

it will be easy to generate collections of points in $B_{\mathbb{E}^2}$ and $B_{\mathbb{S}^2}$. As we anticipated above in the outline of our strategy to estimate $d_{\mathrm{GH}}(B_{\mathbb{E}^2}, B_{\mathbb{H}^2})$, in order to give explicit well-distributed collection of points in $B_{\mathbb{H}^2}$, it is enough to give a well-distributed collection of points in $B_{\mathbb{E}^2}$ and consider its image under the exponential map $\exp_0 : B_{\mathbb{E}^2} \to B_{\mathbb{H}^2}$, where we have identified $B_{\mathbb{E}^2}$ with the ball of radius one of $\mathbb{R}^2 \cong T_0\mathbb{H}^2$, i.e. the tangent space of $\mathbb{H}^2$ at the point 0. However, we should remark that our description of $\mathbb{H}^n$ will not be any of the three most standard ones: namely the *hyperboloid model*, the *the Poincaré ball model*, or the *upper half-plane model*. We describe $\mathbb{H}^n$ as the differentiable manifold $\mathbb{R}^n$ (of Cartesian coordinates $x, y_1, \ldots, y_{n-1}$) equipped with the Riemannian metric $g_{-1} = dx^2 + e^{2x}(dy_1^2 + \cdots dy_{n-1}^2)$. This metric is complete and has constant sectional curvature of -1, which implies that $(\mathbb{R}^n, g_{-1})$ is isometric to $\mathbb{H}^n$. In the hyperboloid model of $\mathbb{H}^2$, we view this space as a submanifold of $\mathbb{R}^3$ (although with a distorted metric, not the one induced from the ambient $\mathbb{R}^3$). In this model of $\mathbb{H}^2$, one can explicitly describe by the following assignment $\exp_0 : \mathbb{R}^2 \to \mathbb{H}^2$ by

$$(x, y) \mapsto \left( \frac{x\sinh(\sqrt{x^2 + y^2})}{\sqrt{x^2 + y^2}}, \frac{y\sinh(\sqrt{x^2 + y^2})}{\sqrt{x^2 + y^2}}, \cosh(\sqrt{x^2 + y^2}) \right). \tag{21}$$

We use this explicit formula of the exponential map with coordinates in the hyperboloid model of $\mathbb{H}^2$, to give a explicit formula for the exponential map with coordinates in the model of $\mathbb{H}^2$ that we introduced above, denoted by $(\mathbb{R}^2, g_{-1})$. In order to change coordinates from the hyperboloid model to the Poincaré ball model, we use the following isometry (which is well-known and can be thought of as a hyperbolic version of the stereographic projection from $\mathbb{S}^2$ to $\mathbb{R}^2$):

$$(x, y, z) \mapsto \left( \frac{x}{1 + z}, \frac{y}{1 + z} \right). \tag{22}$$

To change coordinates from the Poincaré ball model to the upper half-plane model, one can use the following isometry:

$$(x, y) \mapsto \left( \frac{-2y}{(x - 1)^2 + y^2}, \frac{1 - x^2 - y^2}{(x - 1)^2 + y^2} \right). \tag{23}$$

The previous assignment comes from the standard Möbius transformation $z \mapsto \frac{z+i}{iz+1}$ that identifies the Euclidean ball of radius one of $\mathbb{C}$ and the upper half plane of $\mathbb{C}$ as Riemann surfaces (which we give explicitly in this case, although it is known to exist and to be unique by the classical Riemann mapping theorem). Finally, to go from coordinates in the upper half-plane model of $\mathbb{H}^2$ to our model $(\mathbb{R}^2, g_{-1})$, we use the isometry

$$(x, y) \rightarrow (-\log y, x). \tag{24}$$

## B.2 Embedding of $\mathbb{H}^n$ into $\mathbb{E}^{6n-6}$

We want to define an isometric embedding of $B^1_{\mathbb{H}^n}$ into $\mathbb{E}^{6n-6}$ ($F$ from Section 3.2). This higher dimensional space is our candidate to fit in the three geometries $\mathbb{E}^n$, $\mathbb{H}^n$ and $\mathbb{S}^n$ to compute their Hausdorff dimensions as subspaces of $\mathbb{E}^{6n-6}$ and hence estimate their Gromov-Hausdorff distances. Before describing such embedding, we introduce several preliminary auxiliary functions.

Let $\chi(t) = \sin(\pi t) \cdot e^{-\sin^{-2}(\pi t)}$ for non-integer values of $t$. A priori, the inverse of $\sin(0) = 0$ does not make sense but since $\lim_{t \to 0^+} \chi(t) = \lim_{t \to 0^-} \chi(t) = 0$, we can set $\chi(0) = 0$ so it is still continuous. In fact, it is smooth and, in particular, integrable. We can say the same at all points when $t$ is an integer, so we set $\chi(t) = 0$ for all integers $t$ and we obtain a smooth function $\chi$ defined on $\mathbb{R}$.

$$A = \int_0^1 \chi(t) dt. \tag{25}$$

We also define

$$\psi_1(x) = \sqrt{\frac{1}{A} \cdot \int_0^{1+x} \chi(t) \, \mathrm{d}t}, \tag{26}$$

and

$$\psi_2(x) = \sqrt{\frac{1}{A} \cdot \int_0^x \chi(t) \, \mathrm{d}t}. \tag{27}$$

We set $c$ to be the constant

$$c = 2 \max \{G_1, G_2\}, \tag{28}$$

defined in terms of the following

$$G_1 = \left\| \frac{\mathrm{d}}{\mathrm{d}x} (\sinh(x) \cdot \psi_1(x)) \right\|_{L^\infty[-2,2]}, \tag{29}$$

$$G_2 = \left\| \frac{\mathrm{d}}{\mathrm{d}x} (\sinh(x) \cdot \psi_2(x)) \right\|_{L^\infty[-2,2]}, \tag{30}$$

$$h(x, y) = \frac{\sinh(x)}{c} \Big( \psi_1(x) \cos(c \cdot y), \psi_1(x) \sin(c \cdot y), \psi_2(x) \cos(c \cdot y), \psi_2(x) \sin(c \cdot y) \Big), \tag{31}$$

$$\psi(x, y) = \Big( \sinh^{-1}(ye^x), \log(\sqrt{e^{-2x} + y^2}) \Big). \tag{32}$$

We also define $f_0(x, y) = \Big( \int_0^{\sinh^{-1}(ye^x)} \sqrt{1 - \varepsilon(t)^2} \mathrm{d}t, \log(\sqrt{e^{-2x} + y^2}), h(\psi(x, y)) \Big)$,

with $\varepsilon$ being

$$\varepsilon = \frac{G_1^2 + G_2^2}{c^2}. \tag{33}$$

This way, we can set

$$f(x, y_1, \ldots, y_{n-1}) = \frac{1}{\sqrt{n-1}} \left( f_0(x, \sqrt{n-1}\, y_1), \ldots, f_0(x, \sqrt{n-1}\, y_{n-1}) \right).$$

Recall that in our case, we use the function $F(\cdot) = f(n = 2, \cdot)$ to map the set of points $Q$ in $\mathbb{H}^2$ to $\mathbb{R}^6$, and for computing

$$\mathrm{d}_{\mathrm{GH}}(B_{\mathbb{E}^2}, B_{\mathbb{H}^2}) \approx \min_{i,j,k} \mathrm{d}_{\mathrm{H}}^{\mathbb{R}^6, f_k, F}(P_i, Q_j) = \min_{i,j,k} \mathrm{d}_{\mathrm{H}}^{\mathbb{R}^6}(f_k(P_i), F(Q_j)). \tag{34}$$

## B.3 Estimation of the Gromov-Hausdorff distance between the Euclidean and Spherical Spaces

In this section we give the analytical derivation of the Gromov-Hausdorff distance between the Euclidean space $\mathbb{E}^n$ and the Spherical space $\mathbb{S}^n$. We first explain the case $n = 1$, where we can visually understand the situation in a better way because all distances will be measured in $\mathbb{E}^2$. Afterwards, we replicate the same argument for arbitrary $n$. Recall that the Spherical model $\mathbb{S}^n$ can be described as the metric subspace of $\mathbb{E}^{n+1}$ corresponding to the Euclidean sphere of radius one. Hence, as a subspace of $\mathbb{R}^{n+1}$, it corresponds to $\{(x_0, \ldots, x_n) : \sum_{i=0}^{n} x_i^2 = 1\}$.

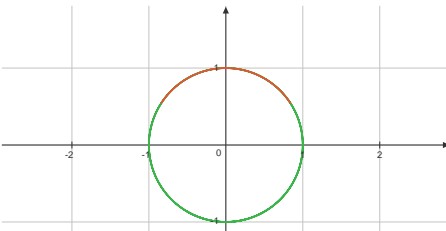

Figure 8: The one-dimensional model of spherical geometry $\mathbb{S}^1$ isometrically embedded in $\mathbb{R}^2$.

The ball of radius one of $\mathbb{S}^1$, denoted by $B_{\mathbb{S}^1}$, is highlighted in red. Our estimation of the Gromov-Hausdorff distance between $\mathbb{E}^1$ and $\mathbb{S}^1$ is motivated by the following observation. In the $y$-axis, the red arc ranges from $\frac{1+\cos(1)}{2}$ to 1. If we consider the following blue segment, representing the ball of

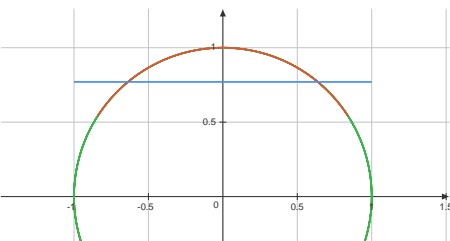

Figure 9: Comparison between $B_{\mathbb{E}^1}$ (in red) and $B_{\mathbb{S}^1}$ (in blue) inside $\mathbb{E}^2$.

radius one of $\mathbb{E}^1$ (denoted by $B_{\mathbb{E}^1}$), we get

$$\sup_{x \in B_{\mathbb{S}^1}} d_{\mathbb{E}^2}(x, B_{\mathbb{E}^1}) = \frac{1 - \cos(1)}{2} \approx 0.23. \tag{35}$$

However, it can be seen that $\sup_{x \in B_{\mathbb{E}^1}} d_{\mathbb{E}^2}(x, B_{\mathbb{S}^1}) = 0.279$. More generally, if the blue line is chosen to be at the height $1 - x$, for some $x$ lying in the closed interval $[0, 1 - \cos(1)]$, it is not hard to see that, for such embedding $f : B_{\mathbb{E}^1} \to \mathbb{E}^2$,

$$dh^{\mathbb{E}^2}(f(B_{\mathbb{E}^1}), B_{\mathbb{S}^1}) = \max \left\{ x, 1 - x, \sqrt{(1 - \sin(1))^2 + (1 - \cos(1) - x)^2} \right\}, \tag{36}$$

whose minimum is attained exactly when $x = \sqrt{(1 - \sin(1))^2 + (1 - x - \cos(1))^2}$, i.e. when

$$x = \frac{(1 - \sin(1))^2 + (1 - \cos(1))^2}{2 - 2\cos(1)} \approx 0.257. \tag{37}$$

This can be seen in the following picture. The two pink lines represent the biggest lengths between

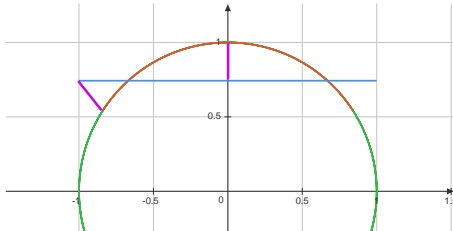

Figure 10: Comparison between $B_{\mathbb{E}^1}$ (in red) and $B_{\mathbb{S}^1}$ (in blue) inside $\mathbb{E}^2$ and a schematic of the biggest lengths (in pink) between the two point sets.

points in $B_{\mathbb{E}^1}$ and $B_{\mathbb{S}^1}$, whose length is approximately equal to 0.257. Since we cannot compute exactly the Gromov-Hausdorff distance $\mathrm{d}_{\mathrm{GH}}(\mathbb{S}^1, \mathbb{E}^1)$, choices have to be made. We have discussed why we would expect it to be somewhere in between 0.23 and 0.257. Since we are considering a very simple embedding for these estimations, it is reasonable to expect $\mathrm{d}_{\mathrm{GH}}(\mathbb{S}^1, \mathbb{E}^1)$ to get closer to the lowest number when considering embeddings of these spaces in more complicated higher dimensional spaces, as the definition of the Gromov-Hausdorff distance allows.

For an arbitrary $n$, we reduce the estimation of $\mathrm{d}_{\mathrm{GH}}(B_{\mathbb{E}^n}, B_{\mathbb{S}^n})$ to the one-dimensional case as follows. Analogously as we did for $n = 1$, given any value of $x$, we can consider $B_{\mathbb{E}^n}$ isometrically embedded in $\mathbb{E}^{n+1}$ by the map $f_x : B_{\mathbb{E}^n} \to \mathbb{R}^{n+1}$ defined by $f((x_0, x_1, \ldots, x_n)) = (x_0, x_1, \ldots, x_n, x)$. We view $B_{\mathbb{S}^n}$ isometrically embedded into $\mathbb{R}^{n+1}$ as the ball of radius one of $\mathbb{S}^n \subset \mathbb{R}^{n+1}$ with centre in the north pole of the sphere, i.e. the point with coordinates $(0, 0, \ldots, 0, 1)$. Given any unit vector $u$ in $\mathbb{R}^{n+1}$ orthogonal to $\vec{n} = (0, 0, \ldots, 0, 1)$, we define $\pi_u$ to be the two-dimensional plane linearly spanned by $u$ and $\vec{n}$. It is clear that, as $u$ ranges over the orthogonal space of $(0, 0, \ldots, 0, 1)$, the intersections $B_{\mathbb{S}^n} \cap \pi_u$ cover the whole $B_{\mathbb{S}^n}$ and the intersections $f_x(B_{\mathbb{E}^n}) \cap \pi_u$ cover the whole $f_x(B_{\mathbb{E}^n})$. Hence, in order to compute $\mathrm{d}_{\mathrm{H}}^{\mathbb{E}^{n+1}}(f_x(B_{\mathbb{E}^n}), B_{\mathbb{S}^n})$, it suffices to compute $\mathrm{d}_{\mathrm{H}}^{\mathbb{E}^{n+1}}(f_x(B_{\mathbb{E}^n}) \cap \pi_u, B_{\mathbb{S}^n} \cap \pi_u)$ for all unit vectors $u$ orthogonal to $\vec{n}$. Moreover, for two such unit vectors $u$ and $v$, there is a rigid motion (a rotation) of the whole ambient space $\mathbb{R}^{n+1}$, that fixes $\vec{n}$ and that maps isometrically $f_x(B_{\mathbb{E}^n}) \cap \pi_u$ to $f_x(B_{\mathbb{E}^n}) \cap \pi_v$ and $B_{\mathbb{S}^n} \cap \pi_u$ to $B_{\mathbb{S}^n} \cap \pi_v$. In particular,

$$\mathrm{d}_{\mathrm{H}}^{\mathbb{E}^{n+1}}(f_x(B_{\mathbb{E}^n}) \cap \pi_u, B_{\mathbb{S}^n} \cap \pi_u) = \mathrm{d}_{\mathrm{H}}^{\mathbb{E}^{n+1}}(f_x(B_{\mathbb{E}^n}) \cap \pi_v, B_{\mathbb{S}^n} \cap \pi_v).$$

So we can do the previous computation for the specific value of $u = (0, 0, \ldots, 1, 0)$ (for clarity, where we mean the unit vector where the $n$-th coordinate is equal to 1 and the rest are zero). Crucially, the projection of $\mathbb{R}^{n+1}$ unto $\mathbb{R}^2$ (by projecting onto the last two coordinates) restrict to isometries on $B_{\mathbb{S}^n} \cap \pi_u$ and $B_{\mathbb{E}^n} \cap \pi_u$, from where it is deduced, analogously as in 36, that, as long as $x \in [0, 1 - \cos(1)]$, we have the following:

$$\mathrm{d}_{\mathrm{H}}^{\mathbb{E}^{n+1}}(f_x(B_{\mathbb{E}^n}) \cap \pi_u, B_{\mathbb{S}^n} \cap \pi_u) = \max\left\{x, 1 - x, \sqrt{(1 - \sin(1))^2 + (1 - \cos(1) - x)^2}\right\}.$$

## B.4 Computational Implementation of the Gromov-Hausdorff Distance between the Remaining Model Spaces of Constant Curvature

In this appendix, we provide further details regarding how the Gromov-Hausdorff distances between the remaining manifolds (between $B_{\mathbb{E}^2}$ and $B_{\mathbb{H}^2}$, and $B_{\mathbb{S}^2}$ and $B_{\mathbb{H}^2}$) were approximated computationally. Note that as discussed in Section 3.3, these distances will be leveraged to compare not only model spaces but product manifolds as well.

### B.4.1 Discretizing the Original Continuous Manifolds

This section discuss how the the points in the corresponding balls of radius one described in Appendix B.1 are generated from a practical perspective. The Euclidean, hyperbolic, and spherical

spaces are continuous manifolds. To proximate the Gromov-Hausdorff distance between them we must discretize the spaces. A more fine-grained discretization results in a better approximation. As previously mentioned in Section 2, to compare $\mathbb{E}^n$, $\mathbb{S}^n$, and $\mathbb{H}^n$, we can take closed balls of radius one in each space. Since these spaces are homogeneous Riemannian manifolds, every ball of radius one is isometric. We can estimate or provide an upper bound for their Gromov-Hausdorff distance by using this method.

In the case of the Euclidean plane, we sample points from the ball, $B_{\mathbb{E}^2} = \{(r\cos(t), r\sin(t)) : r \in [0, 1], t \in [0, 2\pi)\}$, with a discretization of $10,000$ points in both $r$ and $t$. To obtain points in $B_{\mathbb{H}^2}$ we will use the points in $B_{\mathbb{E}^2}$ as a reference, and apply the exponential map, and the change of coordinates described in Appendix B.1 to convert points in $B_{\mathbb{E}^2}$ to points in $B_{\mathbb{H}^2}$. However, due to numerical instabilities instead of sampling from $B_{\mathbb{E}^2}$, we will restrict ourselves to $B'_{\mathbb{E}^2} = \{(r\cos(t), r\sin(t)) : r \in [0.00000001, 0.97], t \in [0, 2\pi)\}$ and use a discretization of $10,000$ as before. Lastly to sample points for the spherical space, $B_{\mathbb{S}^2} = \{(\sin(\beta)\cos(\alpha), \sin(\beta)\sin(\alpha), \cos(\beta)) : \alpha \in [0, 2\pi), \beta \in [0, 1]\}$, we use a discretization of $100$ for both $\alpha$ and $\beta$. The granualirity of the discretization was chosen as a trade-off between resolution and computational time. We observed that the Gromov-Hausdorff distance stabilized for our discretization. However, given the nature of the Gromov-Hausdorff distance, it is difficult to conclude whether some unexpected behaviour could be observed with greater discretization.

### B.4.2 Calculating Constants for the Embedding Function

The next step is to obtain a computational approximation of the mapping function defined in Appendix B.2. The embedding of $\mathbb{H}^n$ into $\mathbb{E}^{6n-6}$ requires computing several constants. Using a standard integral solvers $A \approx 0.141328$, (Equation 25) can be approximated. To calculate the constant $c \approx 10.255014502464228$, we discretize the input $x \in [-2, 2]$ using a step size of $10^{-8}$ to compute $G_1$ and $G_2$ and use a for loop to calculate the max in Equation 28. Note that this requires to constantly reevaluate the integrals for $\psi_1(x)$ and $\psi_2(x)$. Likewise, $G_1$ and $G_2$ are also used to obtain $\varepsilon$ in Equation 33, which is in turn used to calculate the mapping function that maps points in $\mathbb{H}^2$ to the higher dimensional embedding Euclidean space $\mathbb{E}^6$. We use $F$ to map $B_{\mathbb{H}^2}$ to $\mathbb{E}^6$, and we keep those points fixed in space.

### B.4.3 Optimizing the Embedding Functions for the Euclidean and Spherical Spaces

Next to approximate the Gromov-Hausdorff distances:

$$d_{\text{GH}}(B_{\mathbb{E}^2}, B_{\mathbb{H}^2}) \approx \min_{i,j,k} d_{\text{H}}^{\mathbb{R}^6, f_k, F}(P_i^H, Q_j) = \min_{i,j,k} d_{\text{H}}^{\mathbb{R}^6}(f_k(P_i^H), F(Q_j)), \tag{38}$$

and,

$$d_{\text{GH}}(B_{\mathbb{S}^2}, B_{\mathbb{H}^2}) \approx \min_{i,j,k} d_{\text{H}}^{\mathbb{R}^6, g_k, F}(P_i^S, Q_j) = \min_{i,j,k} d_{\text{H}}^{\mathbb{R}^6}(g_k(P_i^S), F(Q_j)), \tag{39}$$

we must optimize $f_k$ and $g_k$. These are the functions used to embed $B_{\mathbb{E}^2}$ and $B_{\mathbb{S}^2}$ in $\mathbb{E}^{6n-6}$. Note that in practice, $B_{\mathbb{E}^2}$ and $B_{\mathbb{S}^2}$ are discretized into $P_i^H$ and $P_i^S$, respectively.

To optimize for $f_k$ we consider all possible permutations of the basis vectors of $\mathbb{E}^6$: $\{e_1, e_2, e_3, e_4, e_5, e_6\}$ for each $f_k$ we consider two elements $\{e_i, e_j\}$ and use those dimensions to embed $\mathbb{E}^2$ in $\mathbb{E}^6$. In principle, we should also consider a small offset given by $F(\mathbf{0}) = \mathbf{0}$, but it is zero regardless. Additionally, during the optimization we also add a small vector (with all entries but a single dimension between zero) to the mapping function to translate the plane in different directions by an offset of between $-0.5$ and $0.5$, with a total of a $100$ steps between these two quantities.

To optimize for $g_k$ we follow a similar procedure in which we consider all permutations of the basis vectors. Note however, that in this case we would have three basis vectors instead of two, given how $B_{\mathbb{S}^2} = \{(\sin(\beta)\cos(\alpha), \sin(\beta)\sin(\alpha), \cos(\beta)) : \alpha \in [0, 2\pi), \beta \in [0, 1]\}$ is sampled. For each permutation family, $P_i^S$, we also consider its negative counterpart, $-P_i^S$, to compute the Gromov-Hausdorff distance as well as experimenting with offsetting the mapping function.

## B.5 Derivation of Number of Product Manifold Combinations in the Search Space

Here, we derive the exact number of nodes in the graph search space in our setting. In particular, we consider the case with three model spaces (the Eucledian plane, the hyperboloid and the hypershere). The growth of the search space can be modelled with growth of a tree, as depicted in Figure 11.

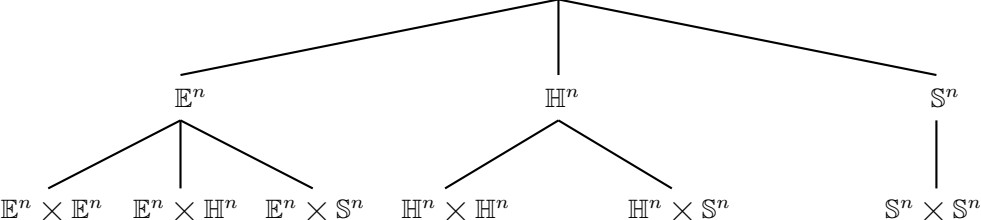

Figure 11: The growth of the graph search space as a function of the number of model spaces used to generate product manifolds can be represented as a tree. Note that as discussed in Section 3.3 we assume commutativity: $\mathcal{M}_i \times \mathcal{M}_j = \mathcal{M}_j \times \mathcal{M}_i$.

To calculate the number of elements in the search space, we define the total number of products at level $h$ of the tree as $N$. We have that $N(h) = N_E(h) + N_H(h) + N_S(h)$, where $N_E(h)$ is the number of Euclidean spaces added to the product manifolds at depth $h$ of the tree, and $N_H(h)$ and $N_S(h)$ represent the same for the hyperboloid and the hypersphere respectively. By recursion, we can write

$$N_E(h) = N_E(h-1) = 1 \tag{40}$$

$$N_H(h) = N_E(h-1) + N_H(h-1) \tag{41}$$
$$= 1 + h \tag{42}$$

$$N_S = N_E(h-1) + N_H(h-1) + N_S(h-1) \tag{43}$$
$$= \frac{h(h+1)}{2} N_E(1) + h N_H(1) + N_S(1) \tag{44}$$
$$= \frac{h(h+1)}{2} + h + 1 \tag{45}$$

hence we have that

$$N(h) = 1 + (1+h) + (1 + h + \frac{h(h+1)}{2}) \tag{46}$$

$$= 3 + \frac{5}{2}h + \frac{1}{2}h^2 \tag{47}$$

To be consistent with the previous notation, we write the above in terms of the number of products $n_p$

$$N(n_p) = 3 + \frac{5}{2}n_p + \frac{1}{2}n_p^2 \tag{48}$$

The total number of nodes in the graph search space for a number of product $n_p$ is then

$$N_T = \sum_{i=1}^{n_p} N(i) \tag{49}$$

## B.6 Visualizing the Graph Search Space

In this appendix, we give a visual depiction of the graph search space we construct for neural latent geometry search. Figure 12 and Figure 13 provide plots of the graph search space as the size of the product manifolds considered increases. For product manifolds composed of a high number of model spaces, visualizing the search space becomes difficult. We omit the strengths of the connections for visual clarity in this plots.

If we focus on Figure 12, we use e, h, and s to refer to the model spaces of constant curvature $\mathbb{E}^2$, $\mathbb{H}^2$, and $\mathbb{S}^2$. In this work, we have considered model spaces of dimensionality two, but the graph search space would be analogous if we were to change the dimensionality of the model spaces. Nodes that include more than one letter, such as hh, ss, ee, etc, refer to product manifolds. For example, $hh$ would correspond to the product manifold $\mathbb{H}^2 \times \mathbb{H}^2$. As discussed in Section 3.3, we can see that connections are only allowed between product manifolds that differ by a single model space. To try to clarify this further, we can see that e, h and s are all interconnected since they only differ by a single model space (one deletion and one addition). Likewise, e is connected to ee, eh, and ee since they only differ by a single model space (one addition). However, e is not connected to hs (one deletion and two additions) nor is ee connected to ss (two deletions and two additions).

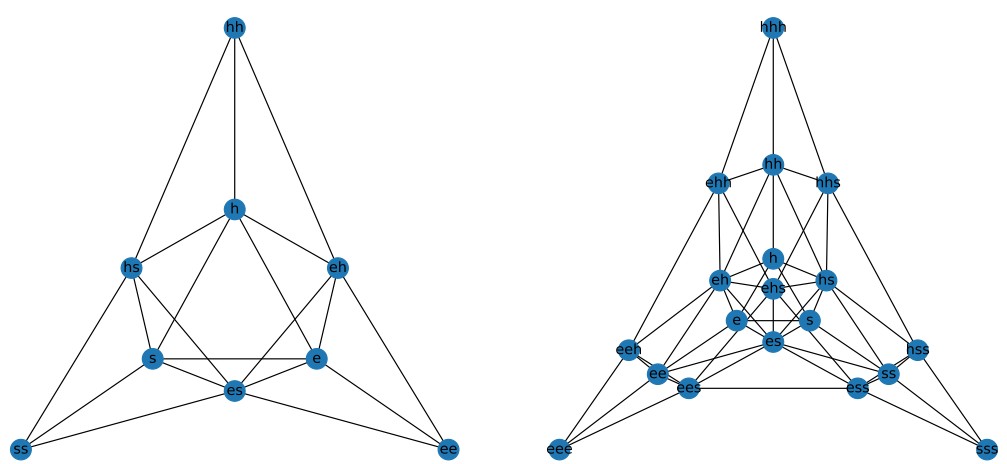

Figure 12: Graph search spaces for $n_p = 2$ (left) and $n_p = 3$ (right). Strength of connectivity is not depicted in the graph.

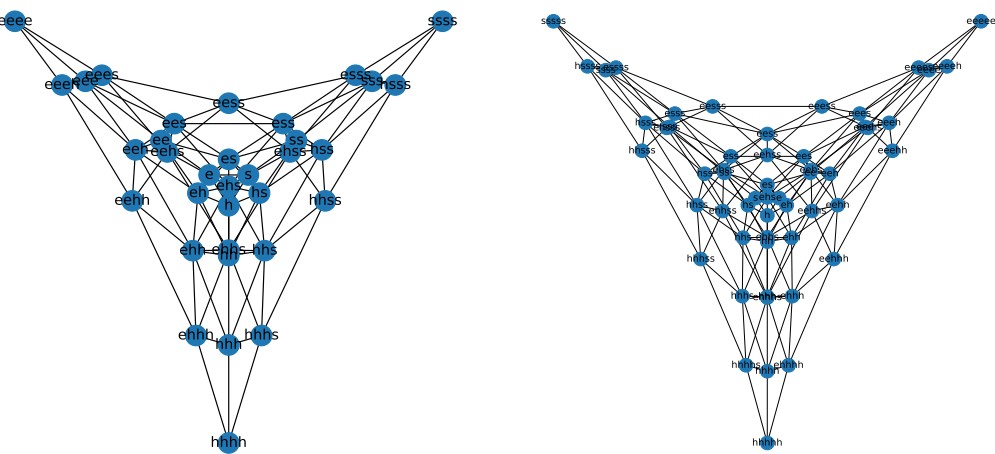

Figure 13: Graph search spaces for $n_p = 4$ (left) and $n_p = 5$ (right). Strength of connectivity is not depicted in the graph.

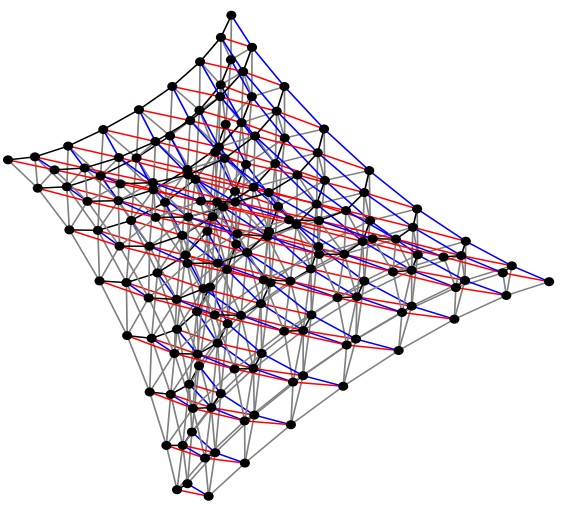

Figure 14: Graph search spaces for $n_p = 8$.

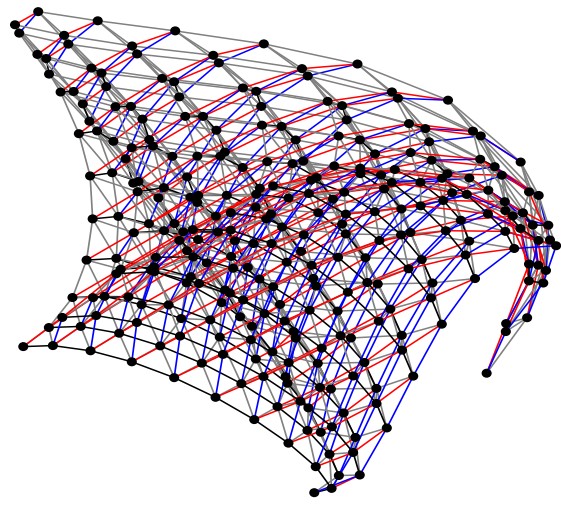

Figure 15: Graph search spaces for $n_p = 10$.

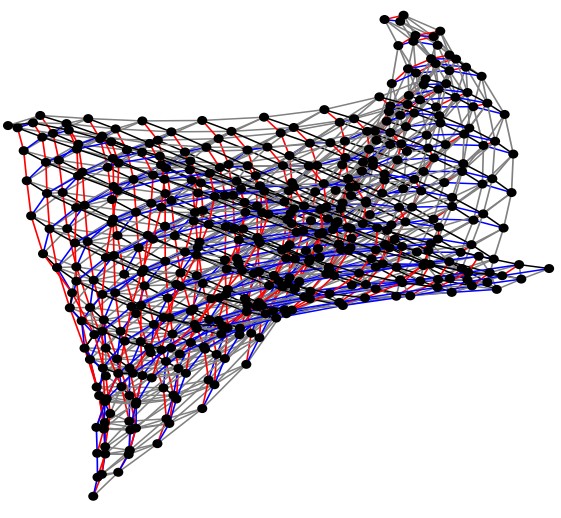

Figure 16: Graph search spaces for $n_p = 12$.

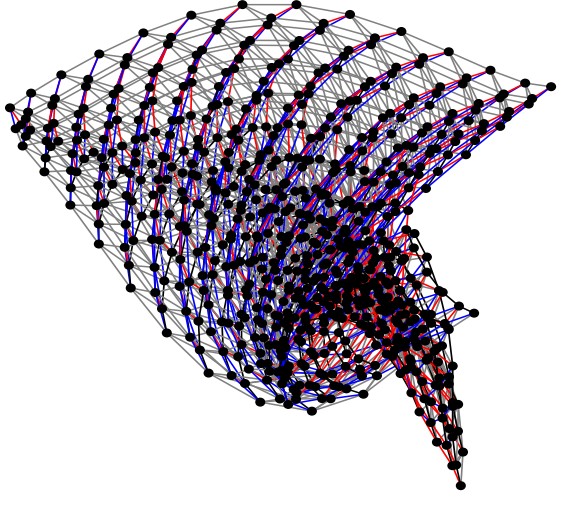

Figure 17: Graph search spaces for $n_p = 14$.

### B.7 Motivating Gromov-Hausdorff Distances for Comparing Latent Geometries

The use of Gromov-Hausdorff distances can be motivated from a theoretical and practical perspective. From a theoretical perspective, the Gromov-Hausdorff distance offers a way to gauge the similarity between metric spaces. This is achieved by casting their representation within a shared space, all the while maintaining the original pairwise distances. In the context of machine learning models, particularly those involving generative models (such as VAEs or GANs), the latent space assumes a pivotal role in shaping the characteristics of the generated data. This also holds true in the case of reconstruction tasks, such as those carried out by autoencoders. Through the application of a metric that factors in their inherent geometries, the aim is to capture the concept of resemblance in data generation, reconstruction, and other downstream tasks which will be heavily affected by the choice of latent geometry.

While traditional metrics like Euclidean or cosine distances might suffice for some cases, they do not always capture the complex and nonlinear relationships between data points in high-dimensional spaces. The Gromov-Hausdorff distance considers the overall structure and shape of these spaces, rather than just individual distances, which can lead to better generalization and robustness. This is especially relevant when dealing with complex data distributions or high-dimensional latent spaces. Moreover, another appealing aspect of Gromov-Hausdorff distance is its invariance to isometric transformations, such as rotations, translations, and reflections. This is a desirable property when comparing latent spaces, as the metric used should reflect the similarity in shape and structure, rather than being influenced by trivial transformations.

While the direct relationship between Gromov-Hausdorff distance and model performance might not be immediately obvious, we argue that if two models have similar latent space geometries, this suggests that they might capture similar underlying data structures, which could lead to similar performance on downstream tasks. This is known as an assumption of *smoothness* in the NAS literature, see Oh et al. [2019]. However, it is important to note that this relationship is not guaranteed, and the effectiveness of Gromov-Hausdorff distance as a proxy measure depends on the specific application and the nature of the models being compared.

### B.8 Method Scalability

The proposed approach sidesteps scalability concerns by preventing the need to recalibrate GH coefficients, as long as one adheres to latent spaces derived from products of model spaces of dimension two. If a higher-dimensional latent space was needed, this could be solved by adding additional "graph slices" with augmented dimensions (as depicted in Figure 2). These slices can be integrated with the lower-dimensional counterpart of the graph search space following the procedure described in the paper. The number of coefficients remains constant at 3, rendering the Gromov-Hausdorff distances free from extra computation overhead when increasing the dimension.

Moreover, in alignment with the manifold hypothesis, data is expected to exist within a low-dimensional manifold. This discourages the exploration of higher dimensions for latent space modeling. It is important to note, nonetheless, that the paper has already delved into relatively large search spaces, which include the exploration of high-dimensional latent spaces.

## C  Background on Bayesian Optimization

Bayesian optimization (BO) is a query-based optimization framework for black-box functions that are expensive to evaluate. It builds a probabilistic model of the objective function using past queries to automate the selection of meta-parameters and minimize computational costs. BO seeks to find the global optimum of a black-box function $\mathfrak{f} : \mathcal{X} \to \mathbb{R}$ by querying a point $\mathbf{x}_n \in \mathcal{X}$ at each iteration $n$ and obtaining the value $y_n = \mathfrak{f}(\mathbf{x}_n) + \varepsilon$, where $\varepsilon \sim \mathcal{N}(0, \sigma^2)$ is the noise in the observation. To do so, BO employs a surrogate to model the function $\mathfrak{f}(\cdot)$ being minimized given the input and output pairs $\mathcal{D}_t = \{(\mathbf{x}_i, y_i = \mathfrak{f}(x_i)\}_{i=1}^N$, and selects the next point to query by maximizing an *acquisition function*. In our work, we use the Gaussian Process (GP)[Williams and Rasmussen, 2006] as the surrogate model and expected improvement (EI)[Mockus et al., 1978] as the acquisition function.

**Preliminaries.** Bayesian optimization (BO) is particularly beneficial in problems for which evaluation is costly, behave as a black box, and for which it is impossible to compute gradients with respect to the loss function. This is the case when tuning the hyperparameters of machine learning models. In our

case, we will use it to find the optimal product manifold signature. Effectively, Bayesian optimization allows us to automate the selection of critical meta-parameters while trying to minimize computational cost. This is done building a probabilistic proxy model for the objective using outcomes recorded in past experiments as training data. More formally, BO tries to find the global optimum of a black-box function $\mathfrak{f} : \mathcal{X} \to \mathbb{R}$. To do this, a point $\mathbf{x}_n \in \mathcal{X}$ is queried at every iteration $n$ and yields the value $y_n = \mathfrak{f}(\mathbf{x}_n) + \varepsilon$, where $\varepsilon \sim \mathcal{N}(0, \sigma^2)$, is a noisy observation of the function evaluation at that point. BO can be phrased using decision theory

$$L(\mathfrak{f}, \{\mathbf{x}_n\}_{n=1}^N) = N \times c + \min_{1 \leqslant n \leqslant N} \mathfrak{f}(\mathbf{x}_n), \tag{50}$$

where $N$ is the maximum number of evaluations, $c$ is the cost incurred by each evaluation, and the loss $L$ is minimized by finding a lower $\mathfrak{f}(\mathbf{x})$. In general, Equation 50 is intractable and the closed-form optimum cannot be found, so heuristic approaches must be used. Instead of directly minimizing the loss function, BO employs a surrogate model to model the function $\mathfrak{f}(\cdot)$ being minimized given the input and output pairs $\mathcal{D}_t = \{(\mathbf{x}_i, y_i = \mathfrak{f}(x_i)\}_{i=1}^N$. Furthermore, in order to select the next point to query, an *acquisition function* is maximized. In this work, the surrogate model is chosen to be a Gaussian Process (GP) [Williams and Rasmussen, 2006] and expected improvement (EI) [Mockus et al., 1978] is used as the acquisition function.

**Gaussian Processes.** A GP is a collection of random variables such that every finite collection of those random variables has a multivariate normal distribution. A GP is fully specified by a *mean function*, $\mu(\mathbf{x})$, and *covariance function* (or *kernel*), denoted as $k(\mathbf{x}, \mathbf{x}')$. The prior over the mean function is typically set to zero, and most of the complexity of the model hence stems from the kernel function. Given $t$ iterations of the optimization algorithm, with an input $\mathbf{x}_t = [x_1, \ldots, x_t]^T$ and output $\mathbf{y}_t = [y_1, \ldots, y_t]^T$, the posterior mean and variance are given by

$$\mu(x_{t+1}|\mathcal{D}_t) = \mathbf{k}(x_{t+1}, \mathbf{x}_t)[\mathbf{K}_{1:t} + \sigma_n^2 \mathbf{I}_t]^{-1} \mathbf{y}_t, \tag{51}$$

and

$$\sigma(x_{t+1}|\mathcal{D}_t) = \mathbf{k}(x_{t+1}, x_{t+1}) - \mathbf{k}(x_{t+1}, \mathbf{x}_t)[\mathbf{K}_{1:t} + \sigma_n^2 \mathbf{I}_t]^{-1} \mathbf{k}(\mathbf{x}_t, x_{t+1}), \tag{52}$$

where we define $[\mathbf{K}_{1:t}]_{i,j} = k(x_i, x_j)$ is the $(i, j)$-th element of the Gram matrix.

**Expected Improvement.** EI is a widely used acquisition function for Bayesian optimization. EI improves the objective function through a greedy heuristic which chooses the point which provides the greatest expected improvement over the current best sample point. EI is calculated as

$$\alpha_{\text{EI}}(\mathbf{x}) = \sigma(\mathbf{x})[\Gamma(\mathbf{x})\Phi(\Gamma(\mathbf{x})) + \mathcal{N}(\Gamma(\mathbf{x})|0, 1)], \tag{53}$$

where

$$\Gamma(\mathbf{x}) = \frac{\mathfrak{f}(\mathbf{x}_{best}) - \mu(\mathbf{x})}{\sigma(\mathbf{x})}, \tag{54}$$

and $\Phi(\cdot)$ denotes the CDF of a standard normal distribution.

## D   Experimental Setup

In this section we provide additional details for the implementation of the experimental setup, including how the datasets are generated, relevant theoretical background, and the hyperparameters used for BO over our discrete graph search space.

### D.1   Synthetic Experiments

To assess the effectiveness of the proposed graph search space, we conduct a series of tests using synthetic data. Our objective is to establish a scenario in which we have control over the latent space and manually construct the optimal latent space product manifold. To achieve this, we initiated the process by generating a random vector, denoted as $\mathbf{x}$, and mapping it to a predetermined "ground truth" product manifold $\mathcal{P}_T$. This mapping is performed using the exponential map for the product manifold $\mathcal{P}_T$, which can be derived based on the model spaces of constant curvature that constitute it. Subsequently, the resulting projected vector is decoded via a neural network with fixed weights, denoted as $f_\theta$, to yield a reference signal $y^{\mathcal{P}_T}$. The rationale behind employing a frozen network is

to introduce a non-linear mapping from the latent space to the signal space. This approach aims to mimic the behavior of a trained network in a downstream task, while disregarding the specific task or model weights involved. By utilizing a frozen network, we can capture the essence of a non-linear mapping without relying on the exact details of the task or specific model weights. The primary goal is to recover this reference signal using the neural latent geometry search (NLGS) approach. To generate the remaining dataset, we employ the same random vector and mapped it to several other product manifolds, denoted as $\{\mathcal{P}_i\}_{i \in n_{\mathcal{P}}}$, comprising an equivalent number of model spaces as $\mathcal{P}_T$ but with distinct signatures. Decoding the projected vector using the same neural network produces a set of signals, denoted as $\{y^{\mathcal{P}_i}\}_{i \in n_{\mathcal{P}}}$. To populate the nodes of the graph search space, we assign the corresponding value of mean squared error (MSE) between $y^{\mathcal{P}_T}$ and $y^{\mathcal{P}_i}$ for each pair. In this manner, our search algorithm aims to identify the node within the graph that minimizes the error, effectively determining the latent geometry capable of recovering the original reference signal. A schematic of the proposed method and specifics on the construction of the network used to decode the signal are shown in Figure 18 and Table 3.

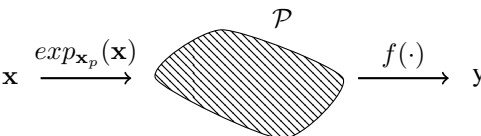

Figure 18: Schematic of procedure used to generate synthetic datasets. We model $f$ as an MLP.

Table 3: Summary of network to generate synthetic datasets.

| Model |
| --- |
| Linear (data dim, 100) - ELU |
| Linear (100, 100) - ELU |
| Linear (100, 100) - ELU |
| Linear (100, 5) - ELU |

## D.2 Autoencoders

Autoencoders are a type of neural network architecture that can be used for unsupervised learning tasks. They are composed of two main parts: an encoder and a decoder. The encoder takes an input and compresses it into a low-dimensional representation, while the decoder takes that representation and generates an output that tries to match the original input. The goal of an autoencoder is to learn a compressed representation of the input data that captures the most important features, and can be used for tasks such as image denoising, dimensionality reduction, and anomaly detection. They have been used in a wide range of applications, including natural language processing, computer vision, and audio analysis.

Table 4: Autoencoder architecture summary.

| Component | Layers |
|---|---|
| **Encoder** | Conv2d (1, 20, 3) - BatchNorm2d - SiLU
Conv2d (20, 20, 3) - BatchNorm2d - SiLU
⋮
(9 repetitions)
⋮
Conv2d (20, 2, 3) - BatchNorm2d - SiLU
Flatten - Linear - SiLU |
| **Decoder** | Linear - SiLU - Unflatten
ConvTranspose2d (2, 20, 3) - BatchNorm2d - SiLU
ConvTranspose2d (20, 20, 3) - BatchNorm2d - SiLU
⋮
(9 repetitions)
⋮
ConvTranspose2d (20, 1, 3) - BatchNorm2d - Sigmoid |

The autoencoder's objective is to learn a compressed representation (latent space) of the input data and use it to reconstruct the original input as accurately as possible. In our experiments, the encoder takes an input image and applies a series of convolutional layers with batch normalization and SiLU activation functions, reducing the image's dimensions while increasing the number of filters or feature maps. The final output of the encoder is a tensor of shape ($\texttt{batchsize}, \texttt{latentdim}, 6, 6$) or ($\texttt{batchsize}, \texttt{latentdim}, 10, 10$), where $\texttt{latentdim}$ is the desired size of the latent space. After flattening this tensor, a fully connected layer is used to map it to the desired latent space size. The encoder's output is a tensor of shape ($\texttt{batchsize}, \texttt{latentdim}$), which represents the compressed representation of the input image. The decoder takes the latent space representation as input and applies a series of transpose convolutional layers with batch normalization and SiLU activation functions, gradually increasing the image's dimensions while decreasing the number of feature maps. The final output of the decoder is an image tensor of the same shape as the input image. The loss functions used for training are the $\texttt{MSELoss}$ (mean squared error) and $\texttt{BCELoss}$ (binary cross-entropy) from the $\texttt{PyTorch}$ library. A summary of the autoencoder is provided in Table 4.

### D.3  Latent Graph Inference

Graph Neural Networks (GNNs) leverage the connectivity structure of graph data to achieve state-of-the-art performance in various applications. Most current GNN architectures assume a fixed topology of the input graph during training. Research has focused on improving diffusion using different types of GNN layers, but discovering an optimal graph topology that can help diffusion has only recently gained attention. In many real-world applications, data may only be accessible as a point cloud of data, making it challenging to access the underlying but unknown graph structure. The majority of Geometric Deep Learning research has relied on human annotators or simplistic pre-processing algorithms to generate the graph structure, and the correct graph may often be suboptimal for the task at hand, which may benefit from rewiring. Latent graph inference refers to the process of inferring the underlying graph structure of data when it is not explicitly available. In many real-world applications, data may only be represented as a point cloud, without any knowledge of the graph structure. However, this does not mean that the data is not intrinsically related, and its connectivity can be utilized to make more accurate predictions.

For these experiments we reproduce the architectures described in Sáez de Ocáriz Borde et al. [2023b], see Table 5 and Table 6 for the original architectures. In our case, we use the GCN-dDGM model leveraging the original input graph inductive bias.

Table 5: Summary of model architectures for experiments for Cora and CiteSeer.

| | | Model | | |
|---|---|---|---|---|
| | | **MLP** | **GCN** | **GCN-dDGM** |
| No. Layer parameters | Activation | Layer type | | |
| | | N/A | N/A | dDGM |
| (No. features, 32) | ELU | Linear | Graph Conv | Graph Conv |
| | | N/A | N/A | dDGM |
| (32, 16) | ELU | Linear | Graph Conv | Graph Conv |
| | | N/A | N/A | dDGM |
| (16, 8) | ELU | Linear | Graph Conv | Graph Conv |
| (8, 8) | ELU | Linear | Linear | Linear |
| (8, 8) | ELU | Linear | Linear | Linear |
| (8, No. classes) | - | Linear | Linear | Linear |

Table 6: dDGM* and dDGM architectures for Cora and CiteSeer.

| No. Layer parameters | Activation | **dDGM*** | **dDGM** |
|---|---|---|---|
| | | Layer type | |
| (No. features, 32) | ELU | Linear | Linear |
| (32, 16 per model space) | ELU | Linear | Graph Conv |
| (16 per model space, 4 per model space) | Sigmoid | Linear | Graph Conv |

### D.3.1 Differentiable Graph Module

In their work, Kazi et al. [2022] presented a general method for learning the latent graph by leveraging the output features of each layer. They also introduced a technique to optimize the parameters responsible for generating the latent graph. The key concept is to use a similarity metric between the latent node features to generate optimal latent graphs for each layer $l$. In this context, $\mathbf{X}^{(0)}$ and $\mathbf{A}^{(0)}$ represent the original input node feature matrix and adjacency matrix, respectively. So that

$$\mathbf{X}^{(0)} = \begin{bmatrix} -\mathbf{x}_1^{(0)}- \\ -\mathbf{x}_2^{(0)}- \\ \vdots \\ -\mathbf{x}_n^{(0)}- \end{bmatrix}, \tag{55}$$

and $\mathbf{A}^{(0)} = \mathbf{A}$ if the adjacency matrix from the dataset, or $\mathbf{A}^{(0)} = \mathbf{I}$ if $\mathcal{G} = (\mathcal{V}, \varnothing)$. The proposed architecture in Kazi et al. [2022] consists of two primary components: the Differentiable Graph Module (DGM) and the Diffusion Module. The Diffusion Module, $\mathbf{X}^{(l+1)} = g_\phi(\mathbf{X}^{(l)}, \mathbf{A}^{(l)})$, may be one (or multiple) standard GNN layers. The first DGM module takes the original node features and connectivity information and produces an updated adjacency matrix

$$\mathbf{X}'^{(l=1)}, \mathbf{A}^{(l=1)} = \text{DGM}(\mathbf{X}^{(0)}, \mathbf{A}^{(0)}). \tag{56}$$

In principle, the DGM module utilizes information from previous layers to generate adjacency matrices at each layer

$$\mathbf{X}'^{(l+1)}, \mathbf{A}^{(l+1)} = \text{DGM}(concat(\mathbf{X}^{(l)}, \mathbf{X}'^{(l)}), \mathbf{A}^{(l)}). \tag{57}$$

To do so, a measure of similarity is used

$$\varphi(\mathbf{x}'^{(l+1)}_i, \mathbf{x}'^{(l+1)}_j) = \varphi(f_\mathbf{\Theta}^{(l)}(\mathbf{x}_i^{(l)}), f_\mathbf{\Theta}^{(l)}(\mathbf{x}_j^{(l)})). \tag{58}$$

In summary, the proposed approach utilizes a parameterized function $f_{\mathbf{\Theta}^{(l)}}$ with learnable parameters to transform node features and a similarity measure $\varphi$ to compare them. The function can be an MLP or composed of GNN layers if connectivity information is available. The output of the similarity measure is used to create a fully-connected weighted adjacency matrix for the continuous

differentiable graph module (cDGM) approach or a sparse and discrete adjacency matrix for the discrete Differentiable Graph Module (dDGM) approach, with the latter being more computationally efficient and recommended by the authors of the DGM paper [Kazi et al., 2022]. To improve the similarity measure $\varphi$ and construct better latent graphs, the approach employs product spaces and Riemannian geometry. Additionally, an extra loss term is used to update the learnable parameters of the dDGM module.

Lastly, we will examine the dDGM module, which utilizes the Gumbel Top-k [Kool et al., 2019] technique to generate a sparse $k$-degree graph by stochastically sampling edges from the probability matrix $\mathbf{P}^{(l)}(\mathbf{X}^{(l)}; \boldsymbol{\Theta}^{(l)}, T)$, which is a stochastic relaxation of the kNN rule, where each entry corresponds to

$$p_{ij}^{(l)} = \exp(-\varphi(T)(\mathbf{x'}_i^{(l+1)}, \mathbf{x'}_j^{(l+1)})). \tag{59}$$

$T$ being a learnable parameter. The primary similarity measure utilized in Kazi et al. [2022] involved computing the distance between the features of two nodes in the graph embedding space

$$p_{ij}^{(l)} = \exp(-T\Delta(\mathbf{x'}_i^{(l+1)}, \mathbf{x'}_j^{(l+1)})) = \exp(-T\Delta(f_{\boldsymbol{\Theta}}^{(l)}(\mathbf{x}_i^{(l)}), f_{\boldsymbol{\Theta}}^{(l)}(\mathbf{x}_j^{(l)}))), \tag{60}$$

where $\Delta(\cdot, \cdot)$ denotes a generic measure of distance between two points. They assumed that the latent features laid in an Euclidean plane of constant curvature $K_{\mathbb{E}} = 0$, so that

$$p_{ij}^{(l)} = \exp(-T\mathfrak{d}_{\mathbb{E}}(f_{\boldsymbol{\Theta}}^{(l)}(\mathbf{x}_i^{(l)}), f_{\boldsymbol{\Theta}}^{(l)}(\mathbf{x}_j^{(l)}))), \tag{61}$$

where $\mathfrak{d}_{\mathbb{E}}$ is the distance in Euclidean space. Based on

$$\text{argsort}(\log(\mathbf{p}_i^{(l)}) - \log(-\log(\mathbf{q}))) \tag{62}$$

where $\mathbf{q} \in \mathbb{R}^N$ is uniform i.i.d in the interval $[0, 1]$, we can sample the edges

$$\mathcal{E}^{(l)}(\mathbf{X}^{(l)}; \boldsymbol{\Theta}^{(l)}, T, k) = \{(i, j_{i,1}), (i, j_{i,2}), ..., (i, j_{i,k}) : i = 1, ..., N\}, \tag{63}$$

$k$ being the number of sampled connections using the Gumbel Top-k trick. The Gumbel Top-k approach utilizes the categorical distribution $\frac{p_{ij}^{(l)}}{\Sigma_r p_{ir}^{(l)}}$ for sampling, and the resulting unweighted adjacency matrix $\mathbf{A}^{(l)}(\mathbf{X}^{(l)}; \boldsymbol{\Theta}^{(l)}, T, k)$ is used to represent $\mathcal{E}(\mathbf{X}^{(l)}; \boldsymbol{\Theta}^{(l)}, T, k)$. It is worth noting that including noise in the edge sampling process can generate random edges in the latent graphs, which can serve as a form of regularization.

Finally, we can summarize a multi-layer GNN using the dDGM module as

$$\mathbf{X'}^{(l+1)} = f_{\boldsymbol{\Theta}}^{(l)}(concat(\mathbf{X}^{(l)}, \mathbf{X'}^{(l)}), \mathbf{A}^{(l)}), \tag{64}$$

$$\mathbf{A}^{(l+1)} \sim \mathbf{P}^{(l)}(\mathbf{X'}^{(l+1)}), \tag{65}$$

$$\mathbf{X}^{(l+1)} = g_\phi(\mathbf{X}^{(l)}, \mathbf{A}^{(l+1)}). \tag{66}$$

Equations 64 and 65 belong to the dDGM module, while Equation 66 is associated with the Diffusion Module. In our study, we extend Equation 59 to measure distances without relying on the assumption utilized in Kazi et al. [2022], which restricts the analysis to fixed-curvature spaces, specifically to Euclidean space where $K_{\mathbb{E}} = 0$.

### D.4 Naive Bayesian Optimization

Naive BO in the main text considers performing BO over a fully-connected unweighted graph. Such a graph is latent geometry agnostic, that is, it does not include any metric geometry inductive bias to provide the optimization algorithm with a sense of closeness between the candidate latent geometries.

# E    Ablation of the Gromov-Hausdorff Coefficients

We evaluate the impact of Gromov-Hausdorff coefficients on the graph search space by conducting an additional set of experiments. We compare the use of an unweighted, pruned graph search space with the introduction of Gromov-Hausdorff coefficients in the datasets considered in this paper. The results are show in Figures 19 to 22 below.

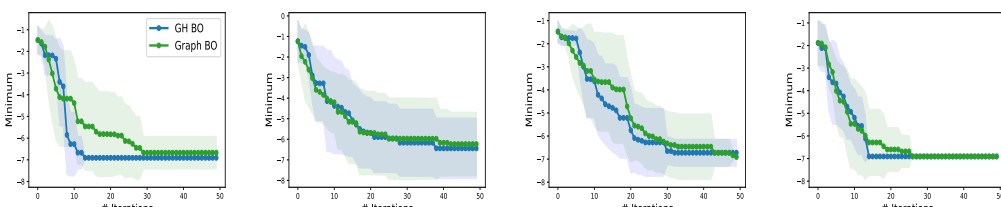

Figure 19: Results (mean and standard deviation over 10 runs) for candidate latent geometries involving product manifolds composed of 13 model spaces. For each plot a different ground truth product manifold $\mathcal{P}_T$ is used to generate the reference signal.

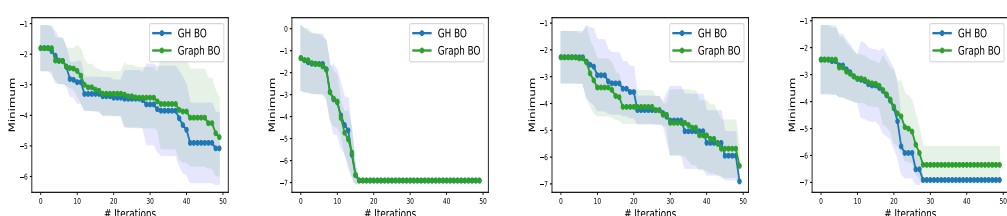

Figure 20: Results (mean and standard deviation over 10 runs) for candidate latent geometries involving product manifolds composed of 15 model spaces. For each plot a different ground truth product manifold $\mathcal{P}_T$ is used to generate the reference signal.

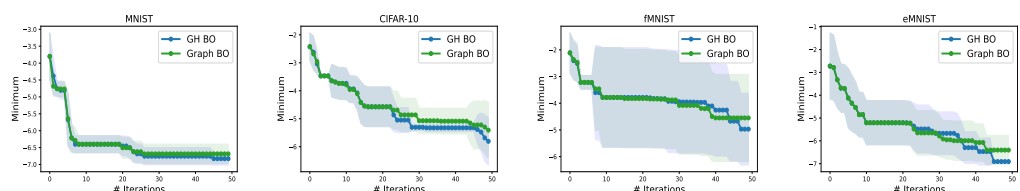

Figure 21: Results (mean and standard deviation over 10 runs) on image reconstruction tasks, for $n_p = 7$.

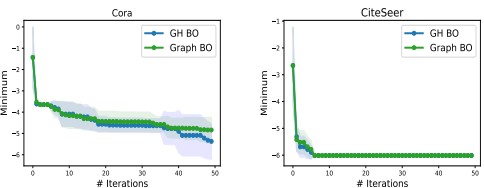

Figure 22: Results (mean and standard deviation over 10 runs) on latent graph inference tasks.

The plots above show that in the majority of cases, the GH weights appear to either match or enhance the performance of the search algorithm. This observation is particularly pronounced

in synthetic tasks, which can be intuitively justified due to the significant impact of changes in the latent geometry on the network's output in such a setup. However, we highlight that a large amount of the performance gains in the algorithm seem to come from the inductive bias endowed to the search algorithm through the graph search space. This could be due to the fact that in image reconstruction tasks, searching for the optimal dimension is more important for reconstruction than finding the optimal latent geometry within a particular dimension. Since the GH coefficients are unity between dimensions, this makes the performance of the two search spaces similar in this context.

It should be noted that setting the inverse of the Gromov-Hausdorff distance as the edges of a search graph is only one potential way to add a geometric inductive bias into the search space. Future work could use similar mechanisms as those developed in this paper but in a different context in order to improve optimization performance. Finally, we would like to highlight that a more thorough analysis of the links between the performance of neural networks and the similarity of their latent geometries is a relevant question that merits further study, which is left for future work.

