# OpenReview forum: "Neural Latent Geometry Search: Product Manifold Inference via Gromov-Hausdorff-Informed Bayesian Optimization"
_NeurIPS.cc/2023/Conference — NeurIPS 2023 poster_

### Official Review · Reviewer_aQg4 · 2023-07-07

**Soundness:** 3 good
**Presentation:** 3 good
**Contribution:** 2 fair
**Rating:** 6
**Confidence:** 2

**Summary:**

This paper addresses the problem of automatically identifying the optimal latent geometry for machine learning models. The neural latent geometry search (NLGS) searches for a latent geometry composed of a product of constant curvature model spaces with minimal query evaluations.The proposed method utilizes the Gromov-Hausdorff distance as a measure to compare and evaluate the distance between different metric spaces. By constructing a graph search space and leveraging Bayesian optimization, the method efficiently searches for the optimal latent geometry.Experimental results on synthetic and real-world datasets demonstrate the effectiveness of the proposed approach in identifying the optimal latent geometry for various machine learning problems.

**Strengths:**

- The authors propose a modeling approach where the latent geometry is represented using the Cartesian product of Riemannian manifolds with constant curvature.
- This paper constructs a graph search space where each node represents a latent space product manifold. The inverse GH distance is used as the weight on the edges, and Bayesian optimization is employed to find the minimum within this space.
- Appropriate figures are included in the paper, effectively illustrating the concepts and results.

**Weaknesses:**

- There is a concern regarding Section 3.1 regarding the potential for overfitting due to the similarity between the objective function $L_{T,A}(g)$ used to determine the latent geometry $g$ and the loss function in the downstream task.
- The paper lacks adequate illustration and explanation for the rationale behind utilizing the Cartesian product of Riemannian manifolds with constant curvature, such as Euclidean space, hyperbolic space, and sphere. It would be beneficial for the authors to provide a more comprehensive insight into the reasons for this choice, including potential advantages like efficient computation or other relevant considerations.
- Although I am not an expert in image reconstruction and latent graph inference, it could be beneficial to compare the GH-BO method with existing benchmarks for these tasks in order to evaluate its performance and assess its effectiveness and competitiveness.

**Other comments:**

It seems that the pdf submitted includes the appendix.

**Questions:**

1. Is it possible to utilize the Gromov-Wasserstein (GW) distance, which is a reformulation of the Gromov-Hausdorff (GH) distance [A], as an alternative for practical computation, given the known computational challenges associated with the GH distance?

2. Why was the diffusion kernel chosen to compute similarity in lines 284-287 instead of using the inverse of the Gromov-Hausdorff distance directly?

3. In the experiment, what does the "Naive BO" method use to be the weight on the fully-connected graph?


[A] Mémoli, Facundo. "Gromov–Wasserstein distances and the metric approach to object matching." *Foundations of computational mathematics* 11 (2011): 417-487.

**Limitations:**

The authors address the limitations of their work. This work does not have a negative societal impact.

---

> ### Author Rebuttal · Authors · 2023-08-09
>
> We thank the reviewer for the positive comments with regard to our work. We hope the additional explanations for some of the weaknesses identified will help to raise the score.
>
> **[Weakness 1]** We thank the reviewer for this comment, but in our opinion, the proposed minimization problem would not necessarily lead to overfitting. The proposed minimization problem has many parallels with the Neural Architecture Search (NAS) problem. In our case, instead of seeking to optimize downstream performance by changing architectural components, we seek to optimize it by changing the latent geometry. Overfitting is prevented by evaluating the performance of the architecture on unseen (validation) data.
>
> **[Weakness 2]** We thank the reviewer for pointing us to this issue. In order to remedy this, we have added a detailed explanation of the motivations behind the use of products of constant curvature model spaces (with ample references to the previous literature) in Appendix A.4, which is included in the overall comment to all reviewers.
>
> **[Weakness 3]** We thank the reviewer for raising this point. We would like to mention that, to the best of our knowledge, there are not benchmarks in the literature focusing on leveraging geometric inductive biases to compare latent geometries in the context of image reconstruction and latent graph inference. Rather, in state-of-the-art papers, most authors focus on manually searching for the most appropriate geometry by exploring different classes of manifolds (mostly product manifolds) to encode the latent space. In our case, we reproduce these state-of-the-art results, and at the same time introduce a principled way to automatically search for the optimal geometry rather than relying on random search. In terms of performance on downstream tasks, we would like to highlight that the use of an advantageous latent geometry has previously led to state-of-the-art performance [2,3].
>
> **[Question 1]** We thank the reviewer for raising this point. In the literature, one can find techniques to estimate both distances, however, it seems that the Gromov-Wasserstein gives better results when it comes to comparing discrete (e.g. finite) objects, as opposed to the continuous infinitesimal objects that we consider in our work (manifolds). In fact, the Gromov-Wasserstein is more technical because it requires endowing your objects with an additional structure: a measure. For the Riemannian manifolds M we consider, a natural choice for this measure would be to assign (to every bounded measurable subset C in M), the volume of C. It is not clear to the authors that the estimation of the corresponding Gromov-Wasserstein distances presents fewer difficulties than the Gromov-Hausdorff distance. Furthermore, given that our manifolds are homogeneous, the estimation of the Gromov-Hausdorff distances is not too bad, and we introduce several simplifications for this task that come from this symmetry (which would be harder to take advantage of when considering other distances).
>
> **[Question 2]** The GH distance is used to generate the graph search space. BO, which utilizes the diffusion kernel, is performed based on the search space, and hence, the GH distance is implicitly utilized by the diffusion kernel (which is a node-wise kernel defined on the graph search space) as an inductive bias when searching through the graph search space.
>
> (Question 3) Naive BO in the main text considers performing BO over a fully-connected unweighted graph (all weights are set to one). Such a graph is latent geometry agnostic, that is, it does not include any metric geometry inductive bias to provide the optimization algorithm with a sense of closeness between the candidate latent geometries. We have included an additional appendix in the paper to explain this further.
>
> **References**
> (see overall comment)

---

> > ### Comment · Reviewer_aQg4 · 2023-08-16
> > **Response to authors**
> >
> > Thank you for your clarifications that effectively address my concerns. Your explanations really help me to understand the manuscript. After thoroughly reviewing all the reviews and rebuttals, I have decided to adjust my score from 4 to 6.

---

### Official Review · Reviewer_1UmR · 2023-07-08

**Soundness:** 3 good
**Presentation:** 3 good
**Contribution:** 3 good
**Rating:** 7
**Confidence:** 3

**Summary:**

The goal of this paper is to learn the geometry of underlying given data points after mapping them into latent space. Rather than learning a general (nonparametric) Riemannian manifold from the embedding data, the paper models this manifold as a product of certain prototypical manifolds with constant curvature (Euclidean, spherical, hyperbolic). Perhaps the idea is that a general nonlinear manifold can be stitched together using patches from these three types of manifolds. The paper describes a way to map points on these prototypical manifolds to their tangent spaces and back, i.e., the inverse exponential and the exponential map. This way one can express an embedded data point as elements of manifolds using their Euclidean coordinates. Patching together of prototype manifolds is done using a graph representation.

With this geometry, the paper derives a framework to fit a latent manifold to the training data. The loss function is built using the Gromov-Hausdorff distance. This requires isometrically embedding the prototypical manifolds into larger Euclidean spaces, and then utilize Euclidean distances to specific Gromov-Hausdorff distances. The search for optimal graph representation of latent space manifold is conducted through Bayesian optimization.

Several experiments on synthetic and real-world datasets are presented in support of this theory.


**Strengths:**

-- Tackles a difficult problem of manifold learning by doing to in the latent space.

-- The idea of representing arbitrary manifolds by a patching of some constant curvature pieces is interesting. (However, one has to show mathematically that this construction covers all or most of the interesting scenarios.)

-- The use of graph representation to patch together the pieces is also interesting and feasible.

-- The experimental results are reasonably demonstrative of the ideas although I have not evaluated them too carefully.

**Weaknesses:**


-- Why is the manifold learning or fitting performed in the latent space and not the original data space. This involves a choice of embedding which could be arbitrary. Perhaps the isometric nature of the embedding allows this arbitrariness to be introduced.

-- Is it not possible to provided a simple minded sketch of the overall framework first using basic mathematics? There are too many symbols introduced right away and that distracts from a quick, preliminary understanding of the paper.

**Questions:**


Please see my comment above.

**Limitations:**

The paper does not discuss any limitation specifically.

---

> ### Author Rebuttal · Authors · 2023-08-09
>
> We thank the reviewer for the positive comments on our work. We hope that the following will address some of the issues raised.
>
> **[Weakness 1]** We thank the reviewer for raising this concern. The reason why the manifold is fitted in the latent space is a consequence of the manifold hypothesis, which operates under the assumption that the data resides in a low-dimensional latent manifold inside the high-dimensional data space. Therefore, we focus on directly modelling the latent space by finding the optimal product of constant curvature model spaces, which is the current state-of-the-art method to model the geometry of the latent space (see [4]).
>
> **[Weakness 2]** In order to improve clarity, we have added a comment in the introduction which points readers to Section 3.1, which provides a high-level sketch of the problem formulation without going into any of the mathematical details. Specifically, the text added is:
>
> *For a high level description of the proposed framework in simple mathematical terms, we recommend skipping to Section 3.1.*
>
> **References**
> (see overall comment)

---

### Official Review · Reviewer_6Vwv · 2023-07-10

**Soundness:** 3 good
**Presentation:** 3 good
**Contribution:** 3 good
**Rating:** 6
**Confidence:** 4

**Summary:**


The paper introduces an approach to estimate the best latent geometry structure for a machine learning model, trained on a data distribution and for a specific task. The approach assumes a product manifold structure of the latent space of subspace of constant curvature, being able to model product spaces of euclidean, spherical and hyperbolic subspaces. The search over candidate geometries is performed by constructing a graph search space where each node in the graph represents a candidate product manifold, associated with the model performance. Edges represent the inverse of the Gromov- Hausdorff distance between candidate spaces. The search is then conducted using Bayesian optimization algorithms. Experiments on synthetic and real scenarios demonstrate that assuming a graph space which models the topology and geometry of the spaces (with the right distances on the edges) entail finding the correct minima and being more efficient in the optimization process.


**Strengths:**



- The paper experimental results highlight the importance of correctly modeling topology and geometry of the latent space in the search for the product latent space that better suits the data and the task at hand, outlining the importance of modeling the geometry in embedding spaces.

- The paper is well written and clear.

- The ideas developed and explained in the paper are relevant to the representation learning and geometric deep learning communities: they could be used for new modeling strategies of latent spaces of NNs, developing constraints, and posing the product space search itself as a learning problem to find the geometry that better represents the data for a given task.


**Weaknesses:**

- The approach seems not very efficient to scale up to real case scenarios due to the requirement of embedding everything  isometrically in a space of 6n-n dimensions when hyperbolic components are considered. Also it is not entirely evident how the approach would scale up in real scenarios, when growing the number of components (subspaces)  and their dimensions considered.

- The experiment primarily shows that modeling the topology and geometry is important to search for the right geometry. While this is a good contribution by itself, in my opinion another great potential outcome of the pape would be interesting in showing  how the resulting product manifold approximates the latent space, to understand if this tells us something about the data, especially for real dataset (e.g. does the MNIST latent space is better modeled by an euclidean product manifold or different ones, how many components better approximate it, etc..)

- It would be good to add some additional works modeling the latent space of NNs as a product manifold, e.g. :

     - *Fumero, Marco, et al. "Learning disentangled representations via product manifold projection." International conference on machine learning. PMLR, 2021.*

     - *Pfau, David, et al. "Disentangling by subspace diffusion." Advances in Neural Information Processing Systems 33 (2020): 17403-17415.*

     - *Zhang, Sharon, Amit Moscovich, and Amit Singer. "Product manifold learning." International Conference on Artificial Intelligence and Statistics. PMLR, 2021.*

**Questions:**


- Does the method scales well with growing dimensions of the latent space or the requirement to embed everything isometrically for hyperbolic latent space is a bottleneck in scaling up everything?

- Can the isometric embedding requirement be relaxed and allow for some bounded distortion in the embedding space? Could this help in reducing the dimensionality constraints on the embedding space?

- Is the latent space search always made post hoc (after having trained a model)? Do you think that the model training would benefit for enforcing a given geometry during training as well?

- The graph pruning strategy (considering models which differ by a single type of latent space and with similar number of subspaces) also loses something in the latent geometry selection strategy? I.e. would it more beneficial in some cases to have two models that differs of more than one component connected in the graph?

- How is the neural network employed to decode the signals in experiment in section 4.1 trained?

- Are the experiments performed always on a single vector projected on the different product manifold geometries? Especially for real case scenario would not be better to work with multiple samples or some averaged statistics of the latent space?

- For real dataset  (e.g. MNIST autoencoder) it is correct to say that the result of the graph search provides the best geometry for the latent space among the one considered? Does this tell something specific about the geometry and topology of the data? What happens if the model is trained from scratch assuming the selected geometry, will the resulting model have better performances?

- The diffusion kernels signature is nearly invariant to isometric transformation of the graph: does this bring any problem in the graph search (e.g. there are cases in which it would be useful to distinguish two nodes with isometrically similar signatures)? How many components (eigenvalues, eigenvectors pairs) are used to approximate the kernel?


**Limitations:**

The limitations are correctly addressed in the manuscript.

---

> ### Author Rebuttal · Authors · 2023-08-09
>
> We thank the reviewer for the positive review. We hope that any outstanding concerns will be addressed with the answers provided below.
>
> **[W1/Q1]** We would like to thank the reviewer for raising this concern. As mentioned in (L1) for Reviewer v4xj, there are no scalability issues associated with the proposed method, as the GH coefficients do not have to be re-computed. If more dimensions are required in the problem considered, one can simply construct additional “graph slices” (see Figure 2) of higher dimension and connect the nodes to the lower dimensional slices using the procedure described in the paper. In either case, the number of coefficients will remain to be 3, and no additional computational cost will be incurred.
>
> Additionally, we would like to mention that we consider the optimal latent geometry, where, under the manifold hypothesis, there is an assumption that the data should lie in a low-dimensional manifold. This would discourage the use of higher dimensions to model the latent space, but this could be done if necessary. It is worthy of mention that relatively large search spaces have already been explored in the paper, which has not led to an explosion in computational requirements.
>
> **[W2]** We thank the reviewer for this positive comment. We do agree that an exploration of the latent space in reconstruction tasks would be a useful study to gain further insight. For example, discussion with other reviewers has motivated the idea of exploring the links between geometric deformations of the latent space results in different neural network outputs. We do note, however, that prior work (see Appendix E of [2]) has already focussed on these ideas. While we do think this point is relevant and well raised, we believe it would be better to thoroughly analyze this idea as a standalone piece, and not distract the reader from the focus of our proposal, which is related to search.
>
> **[W3]** Thank you for pointing us to relevant works that might have been originally overlooked. We have added the rest of recommended citations.
>
> **[Q2]** We have provided the formula to compute the GH coefficient for any arbitrary embedding space dimensionality and hence, there is no need for relaxing the requirements. Practically speaking we apply our algorithm for model spaces of d=2, but we provide the tools to extend the approach to higher-dimensional embedding spaces.
>
> **[Q3]** Thank you for your question. The choice of latent geometry does not have to be made post-hoc. In fact, recent works [2,3] show that there are significant benefits of adopting a correct geometry in the latent space can lead to state-of-the-art performance. The key idea of our method is to introduce the idea of searching for the optimal geometry that will lead to optimal training and better generalization, and propose a potential solution to the problem by leveraging ideas related to product manifolds (see [4]). The search procedure, however, does require training several architectures with different latent geometries, which our method tries to do with a minimal amount of evaluations by virtue of our geometry-informed search space.
>
> **[Q4]** We thank the reviewer for raising this concern. While additional connections could be added to the graph search space, we believe that an overly-connected graph would effectively destroy the topological information, thereby leading to decreased search performance, as is the case in the fully connected (naive) graph search space. The current choice in structure is made such that there are individual “clusters” of nodes in the same dimension, and the algorithm is not encouraged to make large jumps between these dimensional clusters. This choice ensures the graph has a sense of directionality and that search algorithms can leverage local sectors in the search space.
>
> **[Q5]** This setting is supposed to allow us to control the optimal latent geometry, and to gauge how well our method can find it using the output of a neural network as a signal. For this exercise, training of the neural network is not relevant given that our interest is to test the model’s ability to find the (known) optimal geometry. For that reason, the network weights are actually frozen. We have added some additional explanatory diagrams in Appendix D.1 to clarify this, which is included in the additional PDF.
>
> **[Q6]** We thank the reviewer for pointing this out. For each optimal sub-manifold experiment, the random vector is initialized randomly again, which prevents it from being the same one. In real-world experiments, the latent vector will be different at initialization due to the random initialization of network weights.
>
> **[Q7]** (i) Yes, the model would provide the best-performing latent geometry among those available. Under the manifold hypothesis, this would be suggestive of the fact that the data is approximated by a low manifold dimensional structure in the latent space, whose shape is captured best by said latent product manifold. (ii) If the model was trained from scratch with this inductive bias, it could lead to the architecture having improved performance, which could even be state-of-the-art, see [2,3].
>
> **[Q8]** (i) Yes, this is indeed the case. While it is true that the diffusion kernel is in some cases not ideal, it has been previously used effectively in NAS (see [1]), which motivated its use for our problem. As mentioned, this choice could be refined to account for more recent work [5]. However, we would like to highlight that the optimization procedure itself is not the core contribution of our work, which led to this decision.
> (ii) The whole Laplacian is used to approximate the kernel (i.e. all its eigenvalues and eigenvectors)
>
> **References**
> (see overall comment)

---

> > ### Comment · Reviewer_6Vwv · 2023-08-14
> > **Response to rebuttal**
> >
> > I thank the authors for answering my questions and concers.
> >
> >
> >
> > - **[Q2]** Thanks for the answer.  In practice is still not entirely clear to me how costly it is be to scale the method to higher dimensional  model spaces. Can the author elaborate more on the gap to cover to make the approach usable in practice (this is why I was mentioning approximations in my previous comment)? I think this would be useful also concerning future work.
> >
> >
> >
> > - **[Q6]** Thanks for the answer. Concerning real models does this mean that the selection is made at initialization? What about trained models? how the input vector is chosen in that case?

---

> > > ### Author Response · Authors · 2023-08-15
> > > **Response to the reviewer**
> > >
> > > We thank the reviewer for the reply. To elaborate on some of the questions raised:
> > >
> > > **[Q2]** We thank the reviewer for raising this valid point, and apologize for the confusion in the initial reply. The current assumption in this work is that individual model spaces are of dimension 2. This means that for a particular real-world scenario where one wishes to consider a high-dimensional latent space, it would suffice to add additional higher-dimensional "graph slices" (exemplified in Figure 2 in the paper for a product of two model spaces). In this setting, the number of coefficients remains constant at 3, meaning that the Gromov-Hausdorff distances do not have to be computed again when upscaling to higher dimensions. For higher dimensional model spaces, however, this would require the re-computation of the GH coefficients using the procedures described in Appendix B. While it is true that dealing with higher dimensional embeddings could lead to larger computational costs, we highlight the fact that the underlying data manifold is assumed to exist is a low-dimensional space, which discourages the use of excessively high-dimensional model spaces to form the final product. Furthermore, as mentioned in other replies, we would like to emphasize that the main purpose of this work is to bring NLGS to the machine-learning space and provide an initial solution to the problem. From this perspective, we believe that some of the assumptions made (such as that of the dimensionality of the model spaces) are a reasonable starting point, which could be extended in future work.
> > >
> > > **[Q6]** Yes, the selection of the latent geometry needs to be made at the beginning of training and not once the model has been trained. This allows the model to reach better solutions by using the additional geometric inductive bias during training.

---

### Official Review · Reviewer_v4xj · 2023-07-23

**Soundness:** 3 good
**Presentation:** 3 good
**Contribution:** 3 good
**Rating:** 6
**Confidence:** 4

**Summary:**

A variety of recent advances have provided evidence that learning in the
manifold setting (i.e., assuming that high dimensional data lie along
low-dimensional latent manifolds, aka. the manifold hypothesis) is advantageous. However, determining the optimal geometry of the latent space often requires expertise or the use of expensive hyperparameter tuning.
The present work addresses the problem of learning the optimal latent space
geometry for a variety of latent variable models and downstream tasks as well as
provides a formal approach to the topic. To solve the problem, the authors
introduce the construction of a novel graph search space as a basis for applying
Bayesian optimization. The nodes carry information about different latent space
geometries and the edges carry weights from the calculation of the
Gromov-Hausdorff distance between them. As an example to highlight the details of the approach, product manifolds are used. Experiments with toy and real data sets complete the work and provide empirical evidence for the functionality of the approach.

**Strengths:**

I greatly appreciate any attempt to incorporate geometrically sensitive analysis
into machine learning. From my perspective it is advantageous, to assume that
data incorporates structural information and to give models the ability to
address them.

I really like the careful, clear and independent introduction to the theoretical
content; It makes it easy (I would assume even for non-experts) to easily follow the
authors' arguments.

Further, the content is well organized and, with very few exceptions, has no spelling or grammatical flaws. Empirical evidence is accessed through experiments on toy and real world data and the authors make effort to extend the small-scale experimental setting.

**Weaknesses:**

- W1: The theory is presented in a rather narrow setting of product
  manifolds and the authors simplify the problem setting (e.g., see
  Section 3.1) by choosing a rather low dimension (i.e., $d_i=2$) of each of the model spaces and restrict themselves
  to a (very) limited selection of different possible curvatures $K_i \in \lbrace-1,
  0, 1\rbrace$. According to Section B in the appendix, estimation of the
  Gromov-Hausdorff distance in those cases, i.e., in the class of
  product manifolds parameterized by a limited number of signatures, already seems
  laborious. The presented method is very model space specific and seems
  difficult to generalize to arbitrary manifolds. Overall, this impression
  contradicts my expectations of an automatically optimal latent geometry
  recognition when learning any latent space models. However, will there always be the need of expert knowledge in the field of
  differential geometry to first choose a class of manifolds, then derive a
  class-specific Gromov-Hausdorff distance formula (if tractable or derive
  sufficient upper bounds) and finally construct a graph search space?

- W2: In fact, the authors are aware of, that the class of product manifolds are far from
  being as rich as, e.g., the class of arbitrary Riemannian manifolds (cf. l.
  112ff.). Is it possible to quantify how well this class of manifolds works as
  a proxy for arbitrary latent space model classes? In other words, in the case
  of product manifolds as a model class, how severe are the constraints imposed
  when learning a latent space model, e.g., in the setting of variable Bayesian
  inference using a variational auto-encoder model?

- W3: Considering the points (i) I have rather a broad knowledge on the subject
  and (ii) the authors admit that to their knowledge no computational method for
  the mathematical comparison of product manifolds exists (see l. 58), it is
  nevertheless questionable that no related work exists. I would cherish the
  effort to address prior work even it is in the broader sense of latent space
  geometry or addressed to single properties only, e.g., curvature (see Lubold
  et al., Identifying the latent space geometry of network models through analysis of curvature).

- W4: I'm interested in implementation details, in particular how the presented
  geometry identification fits into a neural network pipeline of a latent space
  model, for instance in a variation auto-encoder. A structured description of the
  methodology in form of a code listing as well as code examples for the purpose
  of reproducibility would be desirable.

However, I would like to reiterate my appreciation for the authors' interest in
expanding the field of latent space models from a geometrical point of view. I'm
sure that by working on the deficiencies mentioned, the work will gain in value
and will deliver a valuable contribution to the community.

**Questions:**

- Q1: In Section 4.1, can you give more insight into the "Ground Truth" product manifold $\mathcal{P}_{T}$? Even if this was chosen arbitrarily, what is its dimension?
- Q2: What measure is used for evaluation in the experiments; i.e., what does
  "Minimum" at the y-axis in all plots refer to? And are there any data
  normalization's involved since I identify no changes in the scale of the plots.
- Q3: I would say I got a good intuition for closed balls of radius one in spaces
  of curvature zero or one, at least for dimensions lower 3. What's a good
  intuition for closed balls in spaces of curvature smaller zero? Would be cool
  to have a small section on this, e.g., as part of the appendix. Especially,
  because those balls are used for comparison of model spaces (see l. 137ff.).

- Minor stuff:
    - l. 129 check wording.
    - l. 134 $b_0 \in B$
    - l. 145 $d_{GH}(B,B\times B')$?
    - l. 180 $\mathbb{E}^{6n-6}$ ?
    - l. 199 $d_{GH}(B_{\mathbb{E}²},B_{\mathbb{H²}})$ twice?
    - l. 352 "Figure 6._In"


**Limitations:**

Limitations are addressed (see e.g. Chapter 5) and mainly concern the focus on a
limited class of models, namely product manifolds formed by spaces of constant
curvature. From my perspective there are important aspects that need to be
discussed additionally, e.g., computational limitations in the case of high
dimensional latent spaces, since here the estimation of the Gromov-Hausdorff
distance in combination with the application of Bayesian optimization usually
causes an enormous computational effort.  According to section B.4 in the
appendix, restrictions related to estimating the Gromov-Hausdorff distance for
the (arguably rather simple) model class under consideration are already
implied. In the case of arbitrarily complex Riemannian manifolds, do these
difficulties even lead to intractability?

No evidence was found that the authors addressed potential negative societal impact. Please make the effort to include this mandatory information in your work.

---

> ### Author Rebuttal · Authors · 2023-08-09
>
> We thank the reviewer for appreciating the importance of the topic that we target in our work. We hope that the following responses to the weaknesses raised by the reviewer will merit a raise in the score.
>
> **[W1]** We thank the reviewer for raising this issue. In the same spirit as our previous response (W4) to Reviewer cFVp, we would first like to stress the fact that simplifying assumptions have been made to the search space in order to run a tractable optimization procedure (similar approaches have been adopted in Neural Architecture Search, see [1]) over it to find the best latent geometry out of those available. Our goal with this work is to introduce the idea of NLGS to the machine learning community (which we believe could enrich the AutoML toolbox in a previously unexplored way) and provide an initial solution that could be improved upon by future work. One such example of improvement could be that of curvature learning (first introduced in section 3 in [2]). We believe, however, that the assumptions made are a reasonable starting point that could be built upon in the future.
>
> **[W2]** Thank you for this comment. It is indeed the case that product manifolds are not as rich as the class of arbitrary Riemannian manifolds. To the best of our knowledge, there is no principled way of quantifying how well product manifolds of constant curvature model spaces cover all the possible latent manifolds. However, product manifolds of constant curvature model spaces offer significant advantages in that there exist closed-form solutions for their exponential maps and geodesic distances between points, which is not the case for arbitrary Riemannian manifolds. In the latter case, these quantities must typically be computed numerically or approximated using specialized algorithms. These computations often involve solving differential equations, and depending on the manifold's curvature and geometry, these solutions can be quite complex and may not have closed-form expressions. This is the main reason behind this assumption, which we have detailed more explicitly in Appendix A.4 (shown in the comment to all reviewers).
>
> With regard to the performance of models operating under these assumptions, we highlight that current state-of-the-art approaches in the literature [2,3] employ product manifolds for geometric modelling of the latent space.
>
> **[W3]** We thank the reviewer for bringing this to our attention. We have added the suggested citation and some others ([6,7]) to address prior work in the field. Nevertheless, we highlight that the use of the GH distance to compare latent spaces is introduced for the first time in our work, to the best of our knowledge.
>
> **[W4]** As mentioned in (W2) addressed to Reviewer cFVp, we have added more details related to the architectures in the Appendix (see additional pdf) and provided the code in an anonymous repository.
>
> **[Q1]** We apologize to the reviewer for the lack of clarity regarding this point. Given that the individual model spaces are of dimension two, product manifolds of 13 and 20 model spaces will have dimensions of 26 and 40 respectively.
>
> **[Q2]** Here, minimum refers to the minimum loss encountered by the model thus far. Indeed, all datasets are normalized for the maximum to be at 1 and the minimum to be at 0.
>
> **[Q3]** Essentially, we should think of $B_1^{\mathbb{H}_3}$ as a manifold that is isomorphic to $B_1^{\mathbb{E}^3}$ which is difficult to cover by balls of smaller radius, in fact, it will be coarsely approximated by a finite tree of balls of small radius (the underlying tree being bigger as the chosen radius is smaller).  Furthermore, the philosophy that traveling around $B_1^{\mathbb{H}_3}$ becomes more and more expensive as we leave the origin is made accurate with the consideration of divergence functions: geodesics in $\mathbb{H}^3$ witness exponential divergence. More precisely, there exists a constant $c>1$ such that for every $r>0$ and for every geodesic $\gamma: \mathbb{R}\rightarrow \mathbb{H}^3$ passing through the origin at time $t=0$ it is true that if $\alpha$ is a path that connects $\gamma(-r)$ and $\gamma(r)$ and that lies entirely outside of the ball of radius $d(0, \gamma(r))$, then $\alpha$ must have length at least $c^r$. Again, this is better appreciated when put in comparison with the Euclidean space $\mathbb{E}^3$. Any geodesic (any line) $\gamma: \mathbb{R}\rightarrow \mathbb{E}^3$ passing through the origin at time $t=0$ will have the property that for all $r>0$ there is a path of length $\pi r+\epsilon$, with $\epsilon>0$ arbitrarily small, that lies entirely outside of the ball of radius $r$ with centre at the origin. The length $\pi r+\epsilon$ is linear in $r$, as opposed to the exponential length that is required in hyperbolic spaces.
>
> **[Q4]**
> l. 129 check wording. Done
> l. 134 fixed.
> l. 145 The GH distance between a product manifold with an additional model space and the original model space is 1, the maximum.
> L.180 yes, fixed
> L.199, fixed in parenthesis should be the distance between S^2 and H^2 instead, thanks for pointing it out
> Line 352, fixed
>
> **[Lim1]** Scalability: In this case, we would like to highlight some of the previously mentioned points regarding the necessity of using product manifolds of constant curvature model spaces due to the closed-form solutions of their geodesics and exponential maps. In terms of tractability, it is important to note that given the way that our graph search space is constructed, there is no need to repeatedly compute the GH distance, as it has been pre-computed and can now just be applied to the graph search space. This makes the algorithm highly efficient and scalable. For additional motivation on the use of the GH distance, we refer the reviewer to Appendix B.7 (overall response), which we have added post-review.
>
> **[Lim2]** Societal Impact: We thank the reviewer and have included this discussion int he paper.
>
> **References**
> (see overall comment)

---

> > ### Comment · Reviewer_v4xj · 2023-08-15
> >
> > I thank the authors very much for their response and for the detailed discussion
> > of my questions and concerns.
> >
> > (W1) I agree that the approach is already contributing to the NeurIPS community
> > in the presented introductory version due to the novelty and soundness.
> > Nevertheless the wording should be toned down and simplifications should be made
> > clear. Additionally, please emphasize the works introductory stage as you did in
> > the rebuttal by reviewer cFVp, since the requiring of the current
> > simplifications in order to run, e.g., a tractable optimization procedure
> > overrates automatic ML in my opinion. Thanks for pointing out an interesting
> > improvement opportunity regarding learning curvature.
> >
> > (W2) This is an understandable reason for choosing the evaluation setup and I
> > agree that product manifolds offer a fairly flexible class.
> >
> > (W3) Thank you! Great; it was definitely not in my interest to question the
> > approach's value of novelty!
> >
> > (W4) Cool, this definitely increases reproducibility. Regarding _Experimental
> > Setup: Table 2_, I am a little confused by the presentation. First, why do the
> > terms _Encoder_ and _Decoder_ appear twice; second, what is meant by _(9
> > repetitions)_; third, does the table present one single setup that is used for
> > all experiments or are there multiple models presented?
> >
> > Questions and Limitations are nicely answered!
> >
> > You have clarified things, and I will increase my score correspondingly, under
> > the assumption that these clarifications will make it into the updated version
> > of the paper.
> >
> > Thank you again.

---

> > > ### Author Response · Authors · 2023-08-21
> > > **Response by authors**
> > >
> > > We thank the reviewer for appreciating our work and raising the score.
> > >
> > > As suggested by the reviewer, we will tone down some of the wording associated with the contributions made in the paper, and make our assumptions (as well as the relationship with previous work) more clear. Further, we would like to clarify that the repetition of Encoder/Decoder is a typo, and "9 repetitions" refers to 9 layers made up of the same components as in the row above that statement in the table. Finally, the table which the reviewer refers to (Table 2) presents the autoencoder architecture used for all experiments, which is kept the same.
> > >
> > > We will aim to correct and clarify the above comments for the final version of the paper.

---

### Official Review · Reviewer_cFVp · 2023-07-27

**Soundness:** 3 good
**Presentation:** 3 good
**Contribution:** 3 good
**Rating:** 6
**Confidence:** 3

**Summary:**

In the past few years, there's been an interesting trend in the ML literature where people have proposed non-Euclidean latent space geometries to improve network performance. However, to my knowledge geometries have always been picked on an ad hoc basis or tuned naively as a hyperparameter - the most common being hyperbolic and hyperspherical spaces. This evokes a natural question: is there an efficient way to choose the best latent geometry? The authors formulate this choice as a task on its own: neural latent geometry search (NLGS).

The authors proceed to propose their own method for NLGS. The method chooses a latent geometry from the set of products of up to a fixed number $n_\mathcal{P}$ of manifolds, where $n_\mathcal{P}$ is a hyperparameter. At a high level, the authors metrize the set of product manifolds, use this metric to structure a graph over the set, and then use this graph as a search space for Bayesian optimization.
- The metric is derived from the Gromov-Hausdorff distance between manifolds. The Gromov-Hausdorff distance isometrically embeds two manifolds into a common space ($6n - 6$ dimensional Euclidean space, for this work) and computes the maximum distance from a point on one manifold to the other manifold. Some effort is involved in actually estimating these distance values.
- The authors then create a graph whose vertices are product manifolds and whose edges connect those product manifolds which differ by a single term in the product. The edges are weighted according to the inverse Gromov-Hausdorff distance.
- The authors then proceed to use Bayesian optimization on the resulting graph to pick the latent geometry which minimizes the loss for the problem of interest.

This Gromov-Hausdorff Bayesian optimization (GH BO) method is tested against a naive baseline and Bayesian optimization on a fully connected graph. It outperforms the two methods on both synthetic and real-world experiments.


**Strengths:**

- NLGS is an interesting and well-motivated problem. Based on the broader literature's interest in latent geometries, NLGS could end up being a very impactful framework for model tuning.
  - As a result, GH BO might also end up being a common tool in the hyperparameter search toolbox.
- The theory and machinery used to develop their Gromov-Hausdorff search space is creative and interesting; this is a very original paper.
- The whole paper is well-written and easy to follow. At a glance, even the appendices are a nice read.
- Their approach very clearly empirically outperforms the baselines, which are reasonable in my opinion.


**Weaknesses:**

1. The choice of Gromov-Hausdorff distance as the metric strikes me as a bit arbitrary. Sure, it's *a* way to metrize the model spaces, but I do not think the paper very clearly justifies it as being the right way for the purpose of ML model performance. Why should I expect two models whose latent geometries are closer in Gromov-Hausdorff distance to perform more similarly? As a general principle of optimization, there needs to be some relationship between the objective function and the search space. My concern here is that this relationship is embodied by the structure of the graph (which is discussed for 1 paragraph), rather than the Gromov-Hausdorff weights themselves (which are developed over more than 2 pages + Appendix B).
   - A natural ablation here would be to test BO with the same graph as is used in GH BO algorithm, except with uniform edge weights of 1 instead of the GH weights. This experiment would test whether the GH weights, and not just the choice of edges, are responsible for performance. Thus far the authors have only ablated against a complete graph.

2. The experiments are not at all reproducible.
   - The authors did not describe their network architectures. The appendix only contains a vague description of the autoencoder architecture without details on the number of layers, weights, filters, etc. Even fewer details are available for the other experiments.
   - The authors did not provide their code. Since their method is somewhat involved, this will also hamper adoption by future work.

3. Clearly GH BO performs better than other methods for the models given, but it's not clear whether these are actually *decent models*. As a result, it's unclear to me how much GH BO will help at scale in realistic settings. Clarifying the model architectures used in the experiments might help.

4. On the theoretical side, several simplifications are made (mostly for computional purposes) which make the final algorithm much less interesting:
   - The search space is restricted to products between manifolds of dimension 2.
   - Only unitary (-1, 1) curvature values are used for the non-Euclidean model spaces due to the discrete setup, whereas other works tune the curvature through gradient descent (perhaps that can be done post hoc, but it would be nicer if it were integrated into NLGS).
   - The authors pruned their search graph by removing edges wherein manifold products differ by more than one model space.
   - As a result, they only needed to compute 3 distinct Gromov-Hausdorff distance values, resulting in only 3 distinct graph weights.

I'm open to raising my score if the authors satisfactorily address concerns #1 and #2.

**Questions:**

What is your opinion on my concern #3? Do you think there's reason to expect GH BO will perform at scale?

Can you please clarify this statement regarding other benchmarks: "Furthermore, as there are no existing algorithms to compute the distance between arbitrary product manifolds, we have not included these methods in our benchmark as they would simply converge to random search."
- Can you not metrize arbitrary product manifolds using distances within the graph structure you've designed?
- Why would they converge to random search?

**Limitations:**

The authors identify *constant curvature* model spaces as a limitation. They also point to limits in the field of Bayesian optimization on graphs.

However, I believe some of the simplifications I pointed out in the weaknesses section are also pertinent. It would be helpful to either (1) explicitly raise these as limitations or (2) further justify them as good choices for the algorithm.

---

> ### Author Rebuttal · Authors · 2023-08-09
>
> We thank the reviewer for the very positive comments regarding our work. We also believe that the concept of NLGS could have far-reaching ramifications in the context of model tuning. We hope that the following explanations and changes will be sufficiently convincing to raise the preliminary score.
>
> **[Weakness 1]** We thank the reviewer for this suggestion.
>
> In this work, we operate under the assumption that if two models have similar latent space geometries, they might capture similar underlying data structures, which could lead to similar performance on downstream tasks. This is a similar assumption as that of *smoothness* in the Neural Architecture Search (NAS) community [1], where neural network architectures with few changes in their choice of parameters and layers are assumed to yield similar performance, even though this may not always be the case. As the reviewer correctly points out, this relationship is not guaranteed (either in our case or in the case of general NAS), and the effectiveness of the Gromov-Hausdorff distance as a proxy measure depends on the specific application and the nature of the models being compared. In fact, we believe that a more thorough analysis of how changes in the latent geometry affect the performance of neural networks is a topic of its own right that could be explored in future work as a standalone idea that we could pursue.
>
> To empirically validate this assumption, we have added the ablation suggested by the reviewer, which shows that the Gromov-Haussdorff graph search space outperforms the search space constructed with the same graph but unit coefficients.
>
> In terms of the reasons behind choosing the Gromov-Hausdorf distance over alternative metrics, we highlight that while traditional metrics like Euclidean or cosine distances might suffice for some cases, they do not always capture the complex and nonlinear relationships between data points in high-dimensional spaces. The Gromov-Hausdorff distance considers the overall structure and shape of these spaces, rather than just individual distances, which can lead to better generalization and robustness. Moreover, another appealing aspect of Gromov-Hausdorff distance is its invariance to isometric transformations. This is a desirable property when comparing latent spaces, as the metric used should reflect the similarity in shape and structure, rather than being influenced by trivial transformations. An explanation of the motivations behind the use of the GH distance has been included in Appendix B.7.
>
> Finally, we would like to highlight that the key contribution of this paper is to introduce NLGS to the machine learning community and provide an initial solution (under some assumptions that we believe to be a reasonable starting point) based on *both* the construction of the graph search space and Gromov-Hausdorff coefficients. However, we hope that this initial attempt will be improved upon in future work.
>
> **[Weakness 2]** We agree with the reviewer on this point. To facilitate the reproducibility and adoption of the proposed framework, we have added a more thorough description of the architectures used in the attached pdf and added the code through an anonymous link sent to the ACs.
>
> **[Weakness 3/Question 1]** We have added more details with regard to the models in the Appendix. Further, we highlight that mixed-curvature autoencoders [2] and latent graph inference with product manifolds [3] have achieved state-of-the-art performance in both reconstruction and downstream tasks, respectively.
>
> **[Weakness 4]** We agree with the reviewer that certain simplifying assumptions have been made to the search space. However, we emphasize that this is a necessary step to be able to run an optimization procedure over it, as is the case with Neural Architecture Search (NAS), which motivated our assumptions. Product manifolds are the current state-of-the-art approach [2,3,4] for adding geometric inductive biases to the latent space, so we choose it as our main focus. This could be generalized in future work and an Appendix A.4 has been added to discuss the rationale behind this further. We would like to emphasize that the main contribution of this work is to bring NLGS to the community and provide a potential solution to the problem.  From this perspective, we believe the assumptions being made constitute a reasonable starting point. We hope that future research can explore further along this direction by relaxing some of the assumptions made in this initial approach.
>
> **[Question 2]** Executing search algorithms on a fully-connected graph would effectively mean treating all geometries as equidistant from one another. When applying a search algorithm to a continuum like the real line, the x values possess orientation and interconnectivity. Through discretization, this could be envisioned as a path graph. However, when completely connecting the x-axis, the algorithm loses directional guidance during searches and becomes unable to differentiate between localities or sectors, due to the uniform proximity of all points. Most search algorithms involve some sort of measure of distance between points, which is not available unless it is modified to be compatible with the proposed graph search space, which would imply that each point chosen would essentially be "random". Note that we focus on search space (and not the optimization procedure) which is the main focus of our work, we focus on a single optimization algorithm.
>
> **[Limitation 1]** We will add a small section in the conclusion to take these points into account.
>
> **References**
> (see overall response)

---

> > ### Comment · Reviewer_cFVp · 2023-08-11
> > **Thank you for the rebuttal**
> >
> > Thank you for addressing my questions and concerns.
> >
> > It's cool to see the GH edge weights substantially outperforming unit weights on the first task. Empirically at least, it appears that some amount of *smoothness* (thank you for defining this) applies to the GH search space. As promised, I'll raise my score, conditional on including these results and the architectural details in the final paper, and on releasing the code. I ask, however, that you also include the graph BO baseline in the two experiments in section 4.2.

---

> > > ### Author Response · Authors · 2023-08-15
> > > **Response to the reviewer**
> > >
> > > We thank the reviewer for raising the score. We will carry out the reviewer's requests for the camera-ready version of the paper.

---

### Author Rebuttal · Authors · 2023-08-09

We thank all reviewers for their feedback. We are glad that they acknowledged the novelty of the problem being addressed and the proposed framework. In the overall response, we would like to address some of the concerns raised in multiple reviews, which we have added in different appendices.

**[Motivations of the GH distance] - Appendix B.7**

The Gromov-Hausdorff distance offers a way to gauge the similarity between metric spaces. This is achieved by casting their representation within a shared space, all the while striving to maintain the original pairwise distances. In the context of machine learning models, particularly those involving generative models (such as VAEs or GANs), the latent space assumes a pivotal role in shaping the characteristics of the generated data. This also holds true in the case of reconstruction tasks, such as those carried out by autoencoders. Through the application of a metric that factors in their inherent geometries, the aim is to capture the concept of resemblance in data generation, reconstruction, and other downstream tasks which will be heavily affected by the choice of latent geometry.

While the direct relationship between Gromov-Hausdorff distance and model performance might not be immediately obvious, we argue that if two models have similar latent space geometries, this suggests that they might capture similar underlying data structures, which could lead to similar performance on downstream tasks. However, it is important to note that this relationship is not guaranteed, and the effectiveness of Gromov-Hausdorff distance as a proxy measure depends on the specific application and the nature of the models being compared.

In summary, the selection of the Gromov-Hausdorff distance as a metric for comparing latent spaces in machine learning models originates from the intention to encompass profound structural resemblances that conventional distance metrics might overlook. Although its direct correlation with model performance could be intricate, it offers a systematic approach to evaluate the similarity of latent geometries within a wider framework.

**[Motivation of using Product Manifolds of model spaces] - Appendix A.4**

Most arbitrary Riemannian manifolds do not have closed-form solutions for their exponential maps and geodesic distances between points, as well as for other relevant mathematical notions. The exponential map takes a point on the manifold and exponentiates a tangent vector at that point to produce another point on the manifold. The geodesic distance between two points on a manifold is the shortest path along the manifold connecting those points.

For simple, well-studied manifolds like Euclidean space, hyperbolic space, and spherical space, closed-form solutions for exponential maps and geodesic distances are available. This is because these manifolds have constant curvature, which allows for more straightforward calculations. However, for most other manifolds, the exponential map and geodesic distances must typically be computed numerically or approximated using specialized algorithms. These computations often involve solving differential equations, and depending on the manifold's curvature and geometry, these solutions can be quite complex and may not have closed-form expressions. For certain types of manifolds, like product manifolds where each factor is a well-understood manifold, the computation of exponential maps and geodesic distances can sometimes be simplified due to the separability of the metric. However, scaling these methods to more general, complex manifolds can be challenging and computationally intensive.

**[Scalability of the GH coefficients] Appendix B.8**

The proposed approach sidesteps scalability concerns by obviating the need to recalibrate GH coefficients, as long as one adheres to latent spaces derived from products of model spaces. Should the situation necessitate a higher-dimensional latent space, the solution lies in effortlessly crafting supplementary "graph slices" with augmented dimensions. These slices can be seamlessly integrated with lower-dimensional counterparts following the procedure described in the paper. The count of coefficients remains constant at 3, rendering the Gromov-Hausdorff distances free from extra computation overhead during method upscaling. Moreover, in alignment with the manifold hypothesis, data is anticipated to exist within a low-dimensional manifold. This discourages an inclination towards higher dimensions for latent space modeling, though the possibility persists for situations demanding it. It is important to note, nonetheless, that the paper has already delved into relatively large search spaces.

**References**

[1] Oh, C., Tomczak, J., Gavves, E. and Welling, M., 2019. Combinatorial bayesian optimization using the graph cartesian product. Advances in Neural Information Processing Systems, 32.

[2] Skopek, O., Ganea, O. E., & Bécigneul, G., 2020. Mixed-curvature variational autoencoders. In 8th International Conference on Learning Representations.

[3] de Ocáriz Borde, H.S., Kazi, A., Barbero, F. and Lio, P., 2022, September. Latent graph inference using product manifolds. In The Eleventh International Conference on Learning Representations (ICLR 2023).

[4] Gu, A., Sala, F., Gunel, B. and Ré, C., 2018, September. Learning mixed-curvature representations in product spaces. In International conference on learning representations.

[5] Zhi, Y.C., Ng, Y.C. and Dong, X., 2023. Gaussian processes on graphs via spectral kernel learning. IEEE Transactions on Signal and Information Processing over Networks.

[6] Hauberg, Søren, Oren Freifeld, and Michael Black. "A geometric take on metric learning." Advances in Neural Information Processing Systems 25 (2012).

[7] Arvanitidis, G., Hansen, L.K. and Hauberg, S., 2017. Latent space oddity: on the curvature of deep generative models. arXiv preprint arXiv:1710.11379

---

### Decision · Program_Chairs · 2023-09-21

**Decision:**

Accept (poster)

**Comment:**

This paper addresses a Bayesian optimization for neural latent geometry search. All reviewers agree that the current paper tackles an important problem, well formulating it as NLGS, where the search is carried out using Bayesian optimization algorithms. The authors did a good job in their rebuttal, resulting in an increase in the overall score.  All reviewers feel that the paper has interesting contributions and is deserved for publication.